# Conditional Lagrangian Wasserstein Flow for Time Series Imputation

## Abstract

Time series imputation is important for numerous real-world applications. To overcome the limitations of diffusion model-based imputation methods, e.g., slow convergence in inference, we propose a novel method for time series imputation in this work, called Conditional Lagrangian Wasserstein Flow (CLWF). Following the principle of least action in Lagrangian mechanics, we learn the velocity by minimizing the corresponding kinetic energy. Moreover, to enhance the model's performance, we estimate the gradient of a task-specific potential function using a time-dependent denoising autoencoder and integrate it into the base estimator to reduce the sampling variance. Finally, the proposed method demonstrates competitive performance compared to other state-of-the-art imputation approaches.

## 1. Introduction

Time series imputation is essential for various practical scenarios in many fields, such as transportation, environment, and medical care, etc. Deep learning-based approaches, such as RNNs, VAEs, and GANs, have been proved to be advantageous compared to traditional machine learning methods on various complex real-world multivariate time series analysis tasks (Fortuin et al., 2020). More recently, diffusion models, such as denoising diffusion probabilistic models (DDPMs) (Ho et al., 2020) and score-based generative models (SBGMs) (Song et al., 2020), have gained more and more attention in the field of time series analysis due to their powerful modelling capability (Lin et al., 2023; Meijer & Chen, 2024).

Although many diffusion model-based time series imputation approaches have been proposed and show their advantages compared to conventional deep learning models (Tashiro et al., 2021; Chen et al., 2021; 2023), they are

[1]Anonymous Institution, Anonymous City, Anonymous Region, Anonymous Country. Correspondence to: Anonymous Author <anon.email@domain.com>.

Preliminary work. Under review by the International Conference on Machine Learning (ICML). Do not distribute.

limited to slow convergence or large computational costs. Such limitations may prevent them being applied to real-world applications. To address the aforementioned issues, in this work, we leverage the optimal transport theory (Villani et al., 2009) and Lagrangian mechanics (Arnol'd, 2013) to propose a novel method, called Conditional Lagrangian Wasserstein Flow (CLWF), for fast and accurate time series imputation.

In our method, we treat the multivariate time series imputation task as a conditional optimal transport problem, whereby the random noise is the source distribution, the missing data is the target distribution, and the observed data is the conditional information. To generate new data samples efficiently and accurately, we need to find the shortest path in the probability space according to the optimal transport theory. To this end, we first project the original source and target distributions into the Wasserstein space via sampling mini-batch OT maps. Afterwards, we construct the time-dependent intermediate samples through interpolating the source distribution and target distribution. Then according to the principle of least action in Lagrangian mechanics (Arnol'd, 2013), the optimal velocity function moving the source distribution to the target distribution is learned in a self-supervised manner by minimizing the corresponding kinetic energy. We can solve the model efficiently using flow matching in a simulation-free manner (Lipman et al., 2022; Liu et al., 2022; Albergo & Vanden-Eijnden, 2023; Tong et al., 2023).

To further improve the model's performance, we leverage the denoising affect of the time-dependent denoising autoencoder (TDAE) model which is trained on the observed time series data to estimate the gradient of task-specific potential function. By doing so, combined with the aforementioned flow model, we can formulate a new path sampler to reduce the sampling variances. Furthermore, we can interpret the gradient of the potential function as the control signal from the perspective of stochastic optimal control (SOC) in data generation (Bellman, 1966; Chen et al., 2021; Caluya & Halder, 2021; Berner et al., 2024), Consequently, the sampling procedure can be viewed as a controlled path integral (Zhang & Chen, 2022). We also explain the variance reduction effect of the new sampler using the Rao-Blackwell theorem (Casella & Robert, 1996). Moreover, we propose a resampling technique using the interpolated conditional

samples to enhance the model's imputation performance.

Finally, CLWF is assessed on three real-word and one synthetic time series datasets for validation. The results obtained show that the proposed method achieves competitive performance and admits faster convergence compared with other state-of-the-art time series imputation methods.

The contributions of the paper are summarized as follows:

- We present Conditional Lagrangian Wasserstein Flow, a novel conditional generative framework based on the optimal transport theory and Lagrangian mechanics;

- We develop the efficient training and inference algorithms to solve the time series imputation problem;

- We establish theoretical links between optimal transport, stochastic optimal control and path measures;

- We demonstrate that the proposed method has achieved competitive performance on time series imputation tasks compared to other state-of-the-art methods.

## 2. Preliminaries

In this section, we will succinctly introduce the fundamentals of stochastic differential equations, optimal transport, Shrödinger Bridge, and Lagrangian mechanics.

### 2.1. Stochastic Differential Equations

We treat the data generation task as an initial value problem (IVP), in which $X_0 \in \mathbb{R}^d$ is the initial data (e.g., some random noise) at the initial time $t = 0$, and $X_T \in \mathbb{R}^d$ is target data at the terminal time $t = T$. To solve the IVP, we consider a stochastic differential equation (SDE) defined by a Borel measurable time-dependent drift function $\mu_t : \mathbb{R}^d \times [0, T] \to \mathbb{R}^d$, and a positive Borel measurable time-dependent diffusion function $\sigma_t : [0, T] \to \mathbb{R}^d_{>0}$. Accordingly, the Itô form of the SDE can be described as follows (Oksendal, 2013):

$$\mathrm{d}X_t = \mu_t(X_t, t)\mathrm{d}t + \sigma_t(t)\mathrm{d}W_t, \qquad (1)$$

where $W_t$ is a Brownian motion/Wiener process. Note that when the diffusion term is not considered, the SDE degenerates to an ordinary differential equation (ODE), which is typically easier to solve numerically. Nonetheless, we will use the SDE for theoretical analysis throughout the paper, as it provides a more general framework. Accordingly, The above SDE's associated forward Fokker-Planck Kolmogorov (FPK) equatio (Risken & Frank, 2012) describing the evolution of the marginal density $p_t(X_t)$ reads

$$\frac{\partial p_t}{\partial t} + \boldsymbol{\nabla} \cdot (p_t \mu_t) = \langle D(t), \nabla^2(p_t) \rangle, \qquad (2)$$

where $D(t) := \frac{1}{2}\sigma^\top(t)\sigma(t)$, $\nabla^2$ represents the Hessian operator, and $\langle \cdot, \cdot \rangle := \mathrm{trace}(\cdot^\top, \cdot)$ represents the Frobenius inner product.

In fact, both Eq. (1) and Eq. (2) reveal the system's dynamics and act as the boundary conditions for the optimization problems introduced in later sections, each with a different focus. When the constraint is given by Eq. (1), the formalism is Lagrangian, depicting the movement of each individual particle. In contrast, when the constraint is Eq. (2), the formalism is Eulerian, representing the evolution of the population as a whole.

### 2.2. Optimal Transport

The optimal transport (OT) problem aims to find the optimal transport plans/maps that move the source distribution to the target distribution (Villani et al., 2009; Santambrogio, 2015; Peyré et al., 2019). In the Kantorovich's formulation of the OT problem, the transport costs are minimized with respect to some probabilistic couplings/joint distributions (Villani et al., 2009; Santambrogio, 2015; Peyré et al., 2019). Let $p_0$ and $p_T$ be two Borel probability measures with finite second moments on the space $\Omega \in \mathbb{R}^d$. $\Pi(p_0, p_T)$ denotes a set of transport plans between these two marginals. Then, the Kantorovich's OT problem is defined as follows:

$$\inf_{\pi \in \Pi(p_0, p_T)} \int_{\mathcal{X} \times \mathcal{Y}} \frac{1}{2}\|x - y\|^2 \pi(x, y)\mathrm{d}x\mathrm{d}y, \qquad (3)$$

where $\Pi(p_0, p_T) = \{\pi \in \mathcal{P}(\mathcal{X} \times \mathcal{Y}) : (\pi^x)_{\#}\pi = p_0, (\pi^y)_{\#}\pi = p_T\}$, with $\pi^x$ and $\pi^y$ being two projections of $\mathcal{X} \times \mathcal{Y}$ on $\Omega$. The minimizer of Eq. (3), $\pi^*$, always exists and is referred to as the OT plan.

Note that Eq. (3) can also include an entropy regularization term, the Kullback–Leibler (KL) divergence $D_{\mathrm{KL}}(\pi\|p_0 \otimes p_T)$. This transforms the original OT problem into the entropy-regularized optimal transport (EROT) problem with Eq. (2) serving as the constraint, which frames the transport problem better in terms of convexity and stability (Cuturi, 2013). In particular, from a data generation perspective, $p_0$ is some random initial noise and $p_T$ is the target data distribution, and we can sample the corresponding OT plan in a mini-batch manner (Tong et al., 2023; 2024; Pooladian et al., 2023).

### 2.3. Shrödinger Bridge

The transport problem in Sec. 2.2 can be further viewed from a distribution evolution perspective, which is particularly suitable for developing the flow-based models that model data generation process. For this reason, the Shrödinger Bridge (SB) problem is introduced herein (Léonard, 2012). Assume that $\Omega \in C^1(\mathbb{R}^d \times [0, T])$, $\mathcal{P}(\Omega)$ is a probability path measure on the path space $\Omega$, then the goal of the SB

problem is to find the following optimal path measure:

$$\mathbb{P}^* = \arg\min_{\mathbb{P}\in\mathcal{P}(\Omega)} D_{\mathrm{KL}}(\mathbb{P}\|\mathbb{Q}),$$
$$\text{subject to } \mathbb{P}_0 = q_0 \text{ and } \mathbb{P}_T = q_T, \quad (4)$$

where $D_{\mathrm{KL}}(\mathbb{P}\|\mathbb{Q}) = \begin{cases} \log\left(\frac{d\mathbb{P}}{d\mathbb{Q}}\right)d\mathbb{P}, & \text{if } \mathbb{P} \ll \mathbb{Q}, \\ +\infty, & \text{otherwise,} \end{cases}$ and $\mathbb{Q}$ is a reference path measure, e.g., Brownian motion or Ornstein-Uhlenbeck process. Moreover, the distribution matching problem in Eq. (3) can be reframed as a dynamical SB problem as well (Gushchin et al., 2024; Koshizuka & Sato, 2023; Liu et al., 2024):

$$\arg\min_\theta \mathbb{E}_{p(X_t)}\left[\frac{1}{2}\|\mu_t^\theta(X_t, t)\|^2\right],$$
$$\text{subject to Eq. (1) or Eq. (2),} \quad (5)$$

where $\theta$ is the parameters of the variational drift function $\mu_t$.

### 2.4. Lagrangian Mechanics

In this section, we formulate the data generation problem within the framework of Lagrangian mechanics (Arnol'd, 2013). Let $p_t$ and $\dot{p}_t := dp_t/dt$ be the density and law of the generalized coordinates $X_t$, respectively. Denoting the kinetic energy as $\mathcal{K}(p_t, \dot{p}_t, t)$ and the potential energy as $\mathcal{U}(p_t, t)$, then the corresponding Lagrangian is given by

$$\mathcal{L}(p_t, \dot{p}_t, t) = \mathcal{K}(p_t, \dot{p}_t, t) - \mathcal{U}(p_t). \quad (6)$$

Further, we assume that Eq. (6) is lower semi-continuous (lsc) and strictly convex in $\dot{p}_t$ in the Wasserstein space. Consequently, $\mathcal{K}(x_t, \mu_t, t)$ and $\mathcal{U}(p_t, t)$ are defined as follows, respectively:

$$\mathcal{K}(x_t, \mu_t, t) := \mathbb{E}_{p_t(x_t)}\left[\int_0^T \frac{1}{2}\|\mu_t(x_t, t)\|^2 dt\right], \quad (7)$$

$$\mathcal{U}(p_t, t) := \int_{\mathbb{R}_d} V_t(x_t) p_t(x_t) dx_t, \quad (8)$$

where $V_t(X_t)$ is the potential function. Then the *action* in the context of Lagrangian mechanics is defined as follows:

$$\mathcal{A}(\mu_t(x_s t)) := \int_0^T \int_{\mathbb{R}_d} \mathcal{L}(x_t, \mu_t, t) dx_t dt. \quad (9)$$

According to *the principle of least action* (Feynman, 2005), the shortest path is the one minimizing the action, which is aligned with Eq. (4) in the SB theory as well. Therefore, we can leverage the Lagrangian dynamics to tackle the OT problem for data generation. Moreover, to solve Eq. (6), the corresponding stationary condition, i.e., the

Euler-Lagrangian equation (Arnol'd, 2013), needs to be satisfied:

$$\frac{d}{dt}\frac{\partial}{\partial \dot{p}_t}\mathcal{L}(x_t, \mu_t, t) = \frac{\partial}{\partial p_t}\mathcal{L}(p_t, \dot{p}_t, t), \quad (10)$$

with the boundary conditions: $\frac{dX_t}{dt} = \mu_t$, $p_0 = q_0$, and $p_T = q_T$.

## 3. Conditional Lagrangian Wasserstein Flow for Time Series Imputation

In this section, building on the theory introduced in Sec. 2, we propose Conditional Lagrangian Wasserstein Flow, a novel conditional generative method for time series imputation.

### 3.1. Time Series Imputation

Our goal is to impute the missing time series data points based on the observations. To this end, we adopt a conditionally generative approach for time series imputation in the sample space $\mathbb{R}^{K\times L}$, where $K$ represents the dimension of the multivariate time series and $L$ represents sequence length. In our self-supervised learning approach, the total observed data $x^{\mathrm{obs}} \in \mathbb{R}^{K\times L}$ are partitioned into the imputation target $x^{\mathrm{tar}} := x^{\mathrm{obs}} \odot M^{\mathrm{tar}}$ and the condition $x^{\mathrm{cond}} := x^{\mathrm{obs}} \odot M^{\mathrm{cond}}$, where $\odot$ denotes the Hadamard product, $M^{\mathrm{cond}} \in \mathbb{R}^{K\times L}$ and $M^{\mathrm{tar}} \in \mathbb{R}^{K\times L}$ are the condition and target masks, respectively.

Consequently, the missing data points $x^{\mathrm{tar}}$ can be generated based on the conditions $x^{\mathrm{cond}}$ joint with some uninformative initial distribution $x_0 \in \mathbb{R}^{K\times L}$ (e.g., Gaussian noise) at the initial time $t = 0$. Thereby, the imputation task can be described as: $x^{\mathrm{tar}} \sim p(x^{\mathrm{tar}}|x_0^{\mathrm{in}})$, where $x_0^{\mathrm{in}} := \mathrm{Concatenate}(x^{\mathrm{cond}}, x_0) \in \mathbb{R}^{K\times L\times 2}$ is the the total input of the model.

### 3.2. Interpolation in Wasserstein Space

To solve Eq. (7), we need to sample the intermediate variable $X_t$ in the Wasserstein space first. To do so, the interpolation method is adopted to construct the intermediate samples (Liu et al., 2022; Albergo & Vanden-Eijnden, 2023; Tong et al., 2024). According to the OT and SB problems introduced in Sec. 2, we define the following time-differentiable interpolant:

$$I_t : \Gamma \times \Gamma \to \Gamma \quad \text{such that } I_0 = X_0 \text{ and } I_T = X_T, \quad (11)$$

where $\Gamma \in \mathbb{R}^d$ is the support of the marginals $p_0(X_0)$ and $p_T(X_T)$, as well as the conditional $p(X_t|X_0, X_T, t)$.

To implement Eq. (11), we first independently sample some random noise $X_0 \sim \mathcal{N}(0, \sigma_0^2)$ at the initial time $t = 0$ and the data samples $X_T \sim p(x^{\mathrm{tar}})$ at the terminal time $t = T$,

respectively. Afterwards, the interpolation method is used to construct the intermediate samples $X_t \sim p(X_t|X_0, X_T, t)$, where $t \sim \text{Uniform}(0, T)$. More specifically, we design the following sampling approach:

$$X_t = \frac{t}{T}(X_T + \gamma_t) + (1 - \frac{t}{T})X_0$$
$$+ \alpha(t)\sqrt{\frac{t(T-t)}{T}}\epsilon, \quad t \in [0, T], \quad (12)$$

where $\gamma_t \sim \mathcal{N}(0, \sigma_\gamma^2)$ is some random noise with variance $\sigma_\gamma$ injected to the target data samples to improve the coupling's generalization property, $\alpha(t) \geq 0$ is a time-dependent scalar, and $\epsilon \sim \mathcal{N}(0, \mathbb{I})$.

Note that Eq. (12) can only allow us to generate time-dependent intermediate samples in the Euclidean space but not the Wasserstein space, which can lead to slow convergence as the sampling paths are not straightened. Hence, to address this issue, we can project the samples $X_T$ and $X_0$ in the Wasserstein space before interpolating to strengthen the probability flow. To this end, we leverage the method adopted in (Tong et al., 2023; 2024; Pooladian et al., 2023) to sample the optimal mini-batch OT maps between $X_0$ and $X_T$ first, and perform the interpolations according to Eq. (12) afterwards. Finally, we have the joint variable $x_t^{\text{in}} := (x^{\text{cond}}, x_t)$ as the input for computing the velocity of the Wasserstein flow.

### 3.3. Velocity Estimation via Flow Matching

To estimate the velocity of the Wasserstein flow $\mu_t(X_t, t)$ in Eq. (1), the previous methods that require trajectory simulation for training can result in long convergence time and large computational costs (Chen et al., 2018; Onken et al., 2021). To circumvent the above issue, in this work we adopt a simulation-free learnning strategy based on the OT theory introduce in Sec. 2.2 (Liu et al., 2022; Tong et al., 2023; Albergo & Vanden-Eijnden, 2023), which turns out to be faster and more scalable to large time series datasets.

By drawing mini-batch interpolated samples of the source distribution and target distribution in the Wasserstein space using Eq. (12), we can now model the variational velocity function via a neural network parameterized by $\theta$. Then, according to Eq. (1), the target velocity can be computed as the difference between the source distribution and target distribution. Therefore, the variational velocity function $\mu_\theta(x_t^{\text{in}}, t)$ can be learned trough

$$\arg\min_\theta \int_0^T \int_\mathbb{R} \left\| \frac{dX_t}{dt} - \mu_t^\theta(x_t, t) \right\|_2^2 dx_t dt \quad (13)$$

$$\approx \arg\min_\theta \mathbb{E}\left[ \left\| \frac{x_t^{\text{tar}} - x_0}{T} - \mu_t^\theta(x_t^{\text{in}}, t) \right\|_2^2 \right]. \quad (14)$$

Since Eq. (14) can be solved by drawing mini-batch samples

in the Wasserstein space and performing stochastic gradient descent accordingly, the learning process operates in a simulation-free manner.

Moreover, note that Eq. (13) also obeys the principle of least action introduced in Sec. 2.4 as it minimizes the kinetic energy described in Eq. (7). Therefore, this also indicates that the geodesic that drives the particles from the source distribution to the target distribution in the OT problem described in Sec. 2 is identified, which, as a result, allows us to generate new samples with fewer simulation steps compared to standard diffusion models.

### 3.4. Gradient of Potential Function

So far, we have demonstrated how to leverage the kinetic energy to estimate the velocity in the Lagrangian described by Eq. (6). Apart from this, we can also incorporate the prior knowledge within the task-specific potential energy into the dynamics, which enables us to further improve the data generation performance. To this end, we let $U_t(X_t) : \mathbb{R}^d \times [0, T] \to \mathbb{R}$ be the task-specific potential function depending on the generalized coordinates $X_t$ (Yang & Karniadakis, 2020; Onken et al., 2021; Neklyudov et al., 2023b), and the dynamics (here, we assume that the particle is solely driven by the drift) of the system $v_t(X_t, t)$ yields

$$\frac{dX_t}{dt} = v_t(X_t, t) = -\nabla_x U_t(X_t). \quad (15)$$

Moreover, since the data generation problem in our case can also be interpreted as a stochastic optimal control (SOC) problem (Bellman, 1966; Fleming & Rishel, 2012; Nüsken & Richter, 2021; Zhang & Chen, 2022; Holdijk et al., 2023; Berner et al., 2024), the existence of such $U_t(X_t)$ is guaranteed by Pontryagin's Maximum Principle (PMP) (Evans, 2024). Please refer to Appendix B for further details.

To estimate $v_t(X_t, t)$, according to the Lagrangian in Eq. (6), we assume that the potential function takes the form $U_t(X_t) \approx -\log \mathcal{N}(X_t|\widehat{X}_t, \sigma_p^2)$, where $\widehat{X}_t$ the estimated mean and $\sigma_p^2$ is the pre-defined variance. As a result, the corresponding derivative is $\nabla_x U_t(X_t) = \frac{X_t - \widehat{X}_t}{\sigma_p^2}$. In terms of practical implementation, we parameterize $\nabla_x U(X_t)$ via a time-dependent denoising autoencoder (TDAE). More specifically, we either pre-train or jointly train the TDAE on the intermediate time series data samples $X_t$ generated by Eq. (12). The input is perturbed with noise, while the reconstruction target remains clean, to achieve the denoising effect. Afterwards, the reconstruction discrepancies of the TDAE are used to approximate the variational $v_t^\phi(X_t, t)$ parametrized by $\phi$ depending on the predicted $X_t$:

$$v_t^\phi(X_t, t) = -\frac{s}{\sigma_p^2}\big(X_t - \text{TDAE}(X_t)\big), \quad (16)$$

where $\text{TDAE}(X_t)$ represents the reconstruction of the TAVE model with input $X_t$, $s := s_0 t(T - t)/T$ with $s_0$

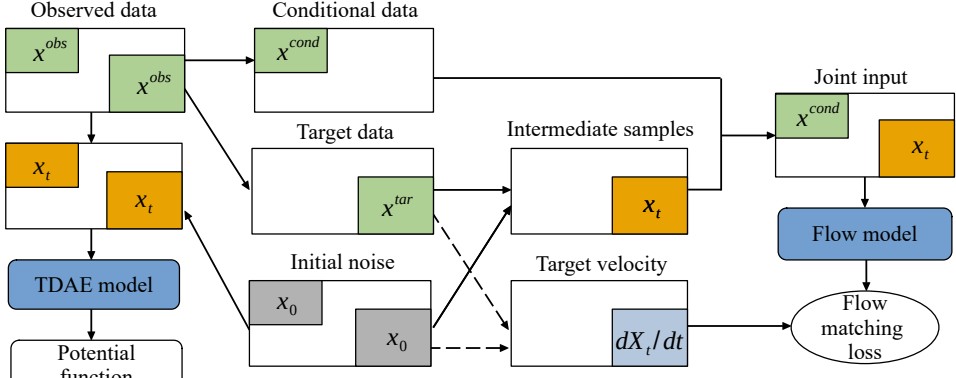

Figure 1: The overall training process of Conditional Lagrangian Wasserstein Flow.

being a positive scalar, and $\sigma_p^2$ is treated as a positive constant for simplicity.

In this manner, we can incorporate the prior knowledge learned from the accessible training data into the sampling procedure established via Eq. (14) to enhance the data generation performance.

### 3.5. Resampling Trick

Note that during inference, the model will generate the new data whose region encompasses both $x_t^{\text{tar}}$ and $x_t^{\text{cond}}$, as the new input for next function evaluation iteration. However, from the problem defined in Sec. 3.1, $x_t^{\text{cond}}$ can be computed accurately by interpolating the condition and the initial noise via Eq. (12). Therefore, we propose the following resampling trick to update the generated intermediate samples $\hat{x}_t$ at time $t$ by stitching the observed data region with the generated data region:

$$\hat{x}_t = \left( \frac{t}{T} x^{\text{obs}} + (1 - \frac{t}{T}) x_0 \right) \odot M^{\text{cond}} + x_t^{\text{gen}} \odot M^{\text{tar}}, \tag{17}$$

where $x_t^{\text{gen}}$ denotes the generated intermediate data samples.

The visualization of the proposed resampling trick can be found in Appendix D.

### 3.6. The Algorithms

We now present the proposed training and inference algorithms. In the training procedure, we minimize the flowing matching loss to learn the variational velocity function $\mu_t^\theta$ using the interpolation method. To estimate the variational drift function $v_t^\phi$, we can calculate the gradient of potential function using the TDVE model. The overall training process of our method is shown in Fig. 1.

In the inference procedure, we use the ODE sampler constructed by $\mu_t^\theta$ to perform the path integral. Moreover, if

---

**Algorithm 1** Training procedure

**Require:** Terminal time: $T$, max epoch, observed data $X^{\text{obs}}$, parameters: $\theta$ and $\phi$.
  **while** epoch $<$ max epoch **do**
    sample $t$, $(x_0, x_T)$
    **if** OT **then**
      sample the mini-batch OT maps;
    **end if**
    sample $x_t$ according to Eq. (12);
    minimize the loss function Eq. (14);
  **end while**
  **if** Rao-Blackwellization **then**
    train a TDAE model using $X^{\text{obs}}$ and $X_0$.
  **end if**

---

we want to further reduce the sampling variances, we can use the drift function $v_t^\phi$ to formulate a new sampler. In addition, we can also choose to use the resampling trick to enhance the data generation performance.

Finally, the detailed training and inference procedures are summarized in Algorithms 1 and 2, respectively.

### 3.7. Discussion

Here, we shed some light on the proposed method's connection to stochastic optimal control, path measures, and Rao-Blackwellization.

**Stochastic optimal control.** We first following the principle of least action in Lagrangian mechanics and optimal transport theory to compute the velocity function $\mu_t$ by minimizing the corresponding kinetic energy. To further improve the data generation performance, we leverage the marginal log density-based potential function to construct the drift function $v_t$. According to the stochastic optimal control theory, it suggests that $v_t$ in fact can act as the opti-

**Algorithm 2** Sampling procedure

**Require:** Step number: $N$, step size: $h_L$,
  sample initial noise $x_0 \sim \mathcal{N}(0, \sigma_0^2)$, conditional information $x^{\text{cond}}$.
  **while** $t < N$ **do**
    $x_t^{\text{in}} = \text{Concatenate}(x^{\text{cond}}, \hat{x}_t)$
    $\hat{x}_{t+1} = \hat{x}_t + \mu_t^\theta(x_t^{\text{in}}, t)\frac{T}{N}$
    **if** Rao-Blackwellization **then**
      $\hat{x}_{t+1} = \hat{x}_{t+1} + v_t^\phi(x_{t+1}^{\text{pred}}, t)\frac{T}{N}$
    **end if**
    **if** Resampling **then**
      $\hat{x}_{t+1} = \left(\frac{(t+1)T}{N}x^{\text{cond}} + \left(T - \frac{(t+1)T}{N}\right)x_0\right) \odot$
      $M^{\text{cond}} + \hat{x}_{t+1} \odot M^{\text{tar}}$
    **end if**
    $t = t + 1$
  **end while**

mal control signal if we consider the data generation process as a controlled SDE. Moreover, the optimal control signal can be attained by solving the corresponding HJB equation using the Hopf-Cole transformation (Evans, 2022) and the FB-SDE theory Anderson (1982); Song et al. (2020). Please refer to Appendix B for the detailed discussion.

**Path measures.** We can now establish the path integral sampler by leveraging the Random-Nikon derivative between the uncontrolled and controlled path measures obtained by the Girsanov theorem (Liptser & Shiryaev, 2013), and obtain the corresponding KL divergence as well. Moreover, we can also derive the ELBO for the marginal density $p(t)$ by solving the associated Fokker-Planck equation using the Feynman-Kac formula (Karatzas & Shreve, 2014). Further details can be found in Appendices B.4 and B.5, respectively.

**Rao-Blackwellization.** If we let the sampler constructed by $\mu_t$ be the based sampler and the sampler constructed by $v_t$ the sufficient statistic, it can be seen that the new sampler can, according to the Rao-Blackwell theorem (Casella & Robert, 1996), improve the data generation performance by reducing the sampling variances. The relevant theoretical details can be found in Appendix C.

## 4. Experiments

In the section, we present the numerical results to demonstrate the effectiveness of our approach.

### 4.1. Datasets

We use one synthetic dataset and three public multivariate time series datasets for validation.

**1) Synthetic dataset** was generated by the function: $x =$

$t\sin(10t + 2\pi\epsilon)$, where $\epsilon \sim \mathcal{N}(0, \mathbb{I})$ and $t \in [0, 1]$ with step size of 0.01. The batch size is 200, the total number of data points is $20,000$, the missing rate of the raw data is $80\%$. $40\%$, $60\%$, and $80\%$ of the datapoints are masked randomly as the imputation targets, denoted as Synthetic 0.4, Synthetic 0.6 and Synthetic 0.8, respectively.

**2) PM 2.5 dataset** (Zheng et al., 2013) was collected from the air quality monitoring sites for 12 months. The missing rate of the raw data is $13\%$. The feature number $K$ is 36 and the sequence length $L$ is 36. In our experiments, only the observed datapoints are masked randomly as the imputation targets.

**3) PhysioNet dataset** (Silva et al., 2012) was collected from the intensive care unit for 48 hours. The feature number $K$ is 35 and the sequence length $L$ is 48. The missing rate of the raw data is $80\%$. $10\%$ and $50\%$ of the datapoints are masked randomly as the imputation targets, denoted as PhysioNet 0.1 and PhysioNet 0.5, respectively.

**4) ETTh1 dataset** (Zhou et al., 2021) was collected from the electric power indicators for 2 years. The feature number $K$ is 24 and the sequence length $L$ is 96. $25\%$, $37.5\%$, and $50\%$ of the datapoints are masked randomly as the imputation targets, denoted as ETTh1 0.25, ETTh1 0.375 and ETTh1 0.5, respectively.

### 4.2. Baselines

For comparison, we select the following state-of-the-art timer series imputation methods as the baselines: 1) GP-VAE (Fortuin et al., 2020), which incorporates the Gaussian Process prior into a VAE model; 2) CSDI (Tashiro et al., 2021), which is based on the conditional diffusion model; 3) CSBI (Chen et al., 2023), which adopts the Schrödinger Bridge diffusion framework; 4) DSPD-GP (Biloš et al., 2023), which combines the diffusion model with the Gaussian Process prior; 5) DLinear (Zeng et al., 2023), which utilizes the moving average kernel for decomposition; 6) LightTS (Zhang et al., 2022), which captures the temporal patterns by continuous and interval sampling; 7) Etsformer (Woo et al., 2022), which proposes to use the exponential smoothing attention and the frequency attention; 8) TimesNet (Wu et al., 2023), which extracts the complex temporal information from the transformed 2D tensors.

### 4.3. Experimental Settings

In terms of architecture choice, both the flow model and the TDAE model are built upon Transformers (Tashiro et al., 2021). We use the ODE sampler for inference and sample the exact optimal transport maps for interpolations to achieve the optimal performance. The optimizer is Adam and the learning rate: 0.001 with linear scheduler. The maximum training epochs is 200. The mini batch size for

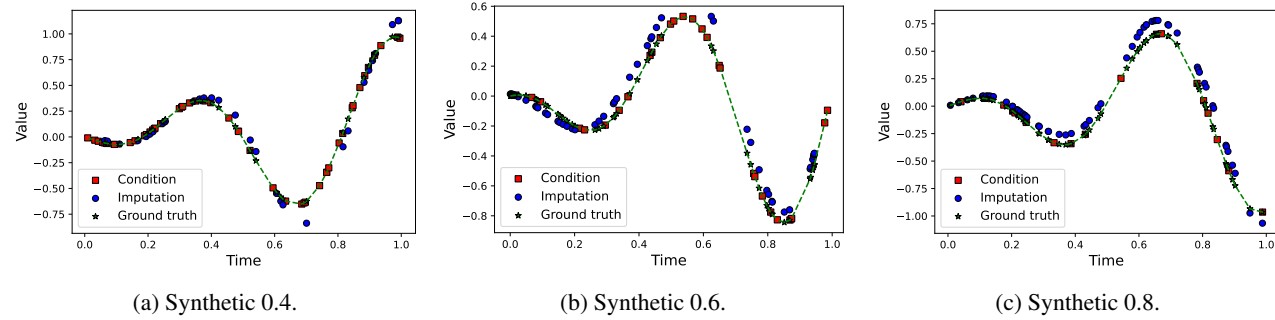

(a) Synthetic 0.4.      (b) Synthetic 0.6.      (c) Synthetic 0.8.

Figure 2: Visualization of the test imputation results on the synthetic data, green dots are the conditions, blue dots are the imputation results, and red dots are the ground truth.

training is 64. The total step number of the Euler method used in CLWF is 15, while the total step numbers for other diffusion models. i.e., CSDI, CSBI, and DSPD-GP are 50, as suggested in their papers. The number of the Monte Carlo samples for inference is 20. The standard deviation $\sigma_0$ for the initial noise $X_0$ is 0.1, and the standard deviation $\sigma_\gamma$ for the injected noise $\gamma_t$ is 0.001. The coefficient $\sigma_p^2$ in the gradient of the potential function is 0.01.

### 4.4. Overall Imputation Results

Tables 1 and 2 show the overall test imputation results on PM 2.5, PhysioNet, and Etth1, respectively. And the results demonstrate that CLWF achieves competitive performance compared with the state-of-the-art methods in terms of RMSE and MAE.

Note that CLWF requires less simulation steps (15) and sampled paths (20) to obtain high-quality data samples, which suggests that CLWF is faster and less computational expensive, compared to other existing diffusion-based time series imputation models, such as CSDI, CSBI, and DSPD.

Table 1: Test imputation results on PM 2.5, PhysioNet 0.1, and PhysioNet 0.5 (5-trial averages). The best are in bold and the second best are underlined.

| Method | PM 2.5 | | PhysioNet 0.1 | | PhysioNet 0.5 | |
|---|---|---|---|---|---|---|
| | RMSE | MAE | RMSE | MAE | RMSE | MAE |
| GP-VAE | 43.1 | 26.4 | 0.73 | 0.42 | 0.76 | 0.47 |
| CSDI | 19.3 | 9.86 | 0.57 | 0.24 | 0.65 | 0.32 |
| CSBI | 19.0 | 9.80 | 0.55 | 0.23 | **0.63** | 0.31 |
| DSPD-GP | 18.3 | **9.70** | 0.54 | 0.22 | 0.68 | 0.30 |
| CLWF | **18.1** | **9.70** | **0.47** | **0.22** | 0.64 | **0.29** |

### 4.5. Ablation Study

To further demonstrate the effectiveness of our proposed method, we conduct the following ablation study experiments.

Table 2: Test imputation results on ETT-h1(5-trial averages). The best are in bold and the second best are underlined.

| Method | ETT-h1 0.25 | | ETT-h1 0.375 | | ETT-h1 0.5 | |
|---|---|---|---|---|---|---|
| | RMSE | MAE | RMSE | MAE | RMSE | MAE |
| DLinear | 0.541 | 0.402 | 0.577 | 0.404 | 0.506 | 0.347 |
| LightTS | 0.469 | 0.347 | 0.544 | 0.382 | 0.463 | 0.318 |
| Etsformer | 0.411 | 0.304 | 0.514 | 0.364 | 0.424 | 0.292 |
| TimesNet | 0.262 | 0.178 | 0.289 | 0.196 | **0.319** | 0.215 |
| CLWF | **0.197** | **0.128** | **0.263** | **0.171** | 0.323 | **0.205** |

**1) Single-path-sample Results.** We compare the test imputation results of CLWF and CSDI by only using one path samples. From the results shown in Table 3, it can be seen that CLWF can achieve relatively good imputation results by only using one single Monte Carlo path integral sample, which indicates that CLWF has smaller sampling variances.

Table 3: Single-sample test imputation results on PM 2.5, PhysioNet 0.1, and PhysioNet 0.5 (5-trial averages).

| Method | PM 2.5 | | PhysioNet 0.1 | | PhysioNet 0.5 | |
|---|---|---|---|---|---|---|
| | RMSE | MAE | RMSE | MAE | RMSE | MAE |
| CSDI | 22.2 | 11.7 | 0.74 | 0.30 | 0.83 | 0.40 |
| CLWF | **18.4** | **10.0** | **0.48** | **0.22** | **0.64** | **0.30** |

**2) Numbers of Diffusion Steps.** We compare the test imputation results of CLWF and CSDI using varying numbers of diffusion steps From the results shown in Table 4, we can see that CLWF has better imputation performance using less simulation steps for inference compared with CSDI, which implies that CLWF has faster convergence during inference.

**3) Effect of Rao-Blackwellization.** We compare the test imputation results of CLWF with and without using the Rao-Blackwellzation, referred to as Base and RB, respectively. From the results shown in Table 5, it can be seen that the new sampler can further improve the time series imputation performance of the base sampler by reducing

Table 4: Test imputation results on PhysioNet 0.1 with different simulation steps (5 trials).

| Method | 5 steps | | 10 steps | | 15 steps | | 20 steps | |
|--------|---------|------|----------|------|----------|------|----------|------|
| | RMSE | MAE | RMSE | MAE | RMSE | MAE | RMSE | MAE |
| CSDI | 0.60 | 0.22 | 0.58 | 0.22 | 0.57 | 0.22 | 0.56 | 0.22 |
| CLWF | **0.48** | **0.22** | **0.47** | **0.22** | **0.47** | **0.22** | **0.48** | **0.22** |

Table 5: Ablation study imputation results on Rao-Blackwellization (5-trial averages).

| Method | PM 2.5 | | PhysioNet 0.1 | | PhysioNet 0.5 | |
|--------|--------|------|---------------|------|---------------|------|
| | RMSE | MAE | RMSE | MAE | RMSE | MAE |
| Base | 18.27 | 9.76 | 0.4802 | **0.2221** | 0.6476 | **0.2991** |
| RB | **18.08** | **9.71** | **0.4785** | 0.2250 | **0.6466** | 0.3003 |

| Method | ETT-h1 0.25 | | ETT-h1 0.375 | | ETT-h1 0.5 | |
|--------|-------------|--------|--------------|--------|------------|--------|
| | RMSE | MAE | RMSE | MAE | RMSE | MAE |
| Base | 0.1999 | 0.1317 | 0.2191 | **0.1422** | 0.2906 | 0.1882 |
| RB | **0.1970** | **0.1266** | **0.2185** | 0.1424 | **0.2891** | **0.1845** |

the variances/RMSEs, which is also supported by the Rao-Blackwell theorem.

Finally, please refer to Appendix E for details on the hardware and software environments used in the experiments, and to Appendix F for additional experimental results.

## 5. Related Work

### 5.1. Diffusion Models

Diffusion models, such as DDPMs (Ho et al., 2020) and SBGM (Song et al., 2020), are considered as the new contenders to GANs on data generation tasks. But they generally take relatively long time to produce high quality samples. To mitigate this problem, the flowing matching methods have been proposed from an OT perspective. For example, ENOT uses the saddle point reformulation of the OT problem to develop a new diffusion model (Gushchin et al., 2024) The flowing matching methods have also been proposed based on the OT theory (Lipman et al., 2022; Liu, 2022; Liu et al., 2023; Albergo & Vanden-Eijnden, 2023; Albergo et al., 2023). In particular, mini-batch couplings are proposed to straighten probability flows for fast inference (Pooladian et al., 2023; Tong et al., 2024; 2023).

The Schrödinger Bridge framework have also been applied to diffusion models for improving the data generation performance of diffusion models. Diffusion Schrödinger Bridge utilizes the Iterative Proportional Fitting (IPF) method to solve the SB problem (De Bortoli et al., 2021). SB-FBSDE proposes to use forward-backward (FB) SDE theory to solve the SB problem through likelihood training (Chen et al., 2022). GSBM formulates a generalized Schrödinger Bridge matching framework by including the task-specific state

costs for various data generation tasks (Liu et al., 2024) NLSB chooses to model the potential function rather than the velocity function to solve the Lagrangian SB problem (Koshizuka & Sato, 2023). Action Matching (Neklyudov et al., 2023a;b) leverages the principle of least action in Lagrangian mechanics to implicitly model the velocity function for trajectory inference. Another classes of diffusion models have also been proposed from an stochastic optimal control perspective by solving the HJB-PDEs (Nüsken & Richter, 2021; Zhang & Chen, 2022; Berner et al., 2024; Liu et al., 2024; Park et al., 2024).

### 5.2. Time Series Imputation

Many diffusion-based models have been recently proposed for time series imputation (Lin et al., 2023; Meijer & Chen, 2024). For instance, CSDI (Tashiro et al., 2021) combines a conditional DDPM with a Transformer model to impute time series data. CSBI (Chen et al., 2023) adopts the FB-SDE theory to train the conditional Schrödinger Bridge model to for probabilistic time series imputation. To model the dynamics of time series from irregular sampled data, DSPD-GP (Biloš et al., 2023) uses a Gaussian process as the noise generator. TDdiff (Kollovieh et al., 2024) utilizes self guidance and learned implicit probability density to improve the time series imputation performance of the diffusion models. However, the time series imputation methods mentioned above exhibit common issues, such as slow convergence, similar to many diffusion models. Therefore, in this work, we proposed CLWF to tackle thess challenges.

## 6. Conclusion

In this work, we proposed CLWF, a novel time series imputation method based on the optimal transport theory and Lagrangian mechanics. To generate the missing time series data, following the principle of least action, CLWF learns a velocity field by minimizing the kinetic energy to move the initial random noise to the target distribution. Moreover, we can also estimate the derivative of a potential function via a TDAE model trained on the observed training data to further improve the performance of the base sampler by Rao-Blackwellization. In contrast with previous diffusion-based models, the proposed requires less simulation steps and Monet Carlo samples to produce high-quality data, which leads to fast inference. For validation, CWLF is assessed on two public datasets and achieves competitive results compared with existing methods.

## Impact Statement

This paper presents work whose goal is to advance the field of time series imputation and deep learning. There are many potential societal consequences of our work, none which we feel must be specifically highlighted here.

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

## A. Notations

Below are some mathematical notations used throughout the paper:

- $\nabla$ denotes the Jacobian operator;

- $\nabla\cdot$ denotes the divergence operator;

- $\nabla^2$ denotes the Hessian operator;

- $\langle\cdot,\cdot\rangle := \text{trace}(\cdot^\top,\cdot)$ denotes the Frobenius inner product.

## B. Stochastic Optimal Control

In this section, we show how the stochastic optimal control theory is related to our data generation task.

### B.1. Cost Function

An SDE controlled by the deterministic control function $u \in \mathbb{U} \subset C(\mathbb{R}^d \times [0, T]; \mathbb{R})$, where $\mathbb{U}$ is a set of admissible controls, reads

$$\mathrm{d}X_t^u = (a + \sigma u)(X_t^u, t)\mathrm{d}t + \sigma(X_t^u, t)\mathrm{d}\overline{W}_t, \tag{18}$$

where $a \in C^1(\mathbb{R}^d \times [0, T]; \mathbb{R}^d)$ is the drift/advection function, $\sigma(t) \in C^1(\mathbb{R}^d \times [0, T]); \mathbb{R}^{d\times d}$ is the diffusion function, and $\overline{W}_t$ is the standard Brownian motion.

Therefore, the data generation task can also be interpreted as a stochastic optimal control (SOC) problem (Bellman, 1966; Fleming & Rishel, 2012; Nüsken & Richter, 2021; Zhang & Chen, 2022; Holdijk et al., 2023; Koshizuka & Sato, 2023; Domingo-Enrich et al., 2023; Berner et al., 2024) whose cost functional $\mathcal{J}$ is defined as:

$$\mathcal{J}(u; x_{\text{init}}, t) = \mathbb{E}\left[\int_t^T h(X_s^u, u, s)\mathrm{d}s + g(X_T^u)\Big| X_t^u = x_{\text{init}}\right], \tag{19}$$

where $h(X_s^u, u, s) := f(X_s^u, s) + \frac{1}{2}\|u(X_s^u, s)\|_2^2$, where $f \in C^1(\mathbb{R}^d \times [t, T]; [0, \infty))$, is the instantaneous/running cost, and $g \in C^1(\mathbb{R}^d; \mathbb{R})$ denotes the terminal cost. The above SOC problem can be solved via dynamic programming (Bellman, 1966; Bertsekas, 2012).

### B.2. Pontryagin's Maximum Principle

Consider the following optimization problem derived from Eq. 19:

$$\arg\min_{u\in\mathcal{U}}\mathcal{J}(u; x_0, t) = \arg\min_{u\in\mathcal{U}}\left\{\int_0^T h(x, u, t)\mathrm{d}t + g(X_T)\right\} \tag{20}$$

$$\text{subject to } \begin{cases} \dot{x}(t) = v(x, u, t), & 0 < t \le T, \\ x(0) = x_0, \end{cases} \tag{21}$$

where $\dot{x}(t) := \frac{\mathrm{d}x(t)}{\mathrm{d}t}$ and $v(x, u, t)$ denotes the system dynamics. Accordingly, the associated Lagrangian functional is defined as

$$\mathcal{L}(x, \lambda, u, t) := \int_0^T \left\{h(x, u, t) + \lambda(t)(\dot{x}(t) - v(x, u, t))\right\}\mathrm{d}t + g(X_T), \tag{22}$$

where $\lambda(t)$ is the Lagrangian multiplier.

Now, let's consider a dynamic system defined by the following Hamiltonian

$$\mathcal{H}(x, \lambda, u) := \mathcal{K} + \mathcal{U}$$
$$= \lambda^\top(t)v(x, u, t) - h(x, u, t) \tag{23}$$

where $\mathcal{K}$ is the kinetic energy, $\mathcal{U}$ is the potential energy, and $\lambda(t)$ serves as the momentum/costate here. Then, the Lagrangian functional in Eq. 22 becomes

$$\mathcal{L}(x, \lambda, u, t) := \int_0^T \left\{ -\mathcal{H}(x, \lambda, u) + \lambda(t)\dot{x}(t) \right\} \mathrm{d}t + g(X_T), \tag{24}$$

We now apply variations of calculus to Eq. (24) by assuming $(x^*(t), \lambda(t), u^*(t))$ is its minimizer and adding perturbation $\delta x, \delta u, \delta \lambda$ with $\delta x(0) = 0$. Now we perform the first-order Taylor expansion:

$$\delta\mathcal{L} := \mathcal{L}(x^* + \delta x, \lambda + \delta\lambda, u^* + \delta u) - \mathcal{L}(x^*, \lambda, u^*)$$

$$\approx \int_0^T \left\{ -\mathcal{H}_x \delta x - \mathcal{H}_\lambda \delta\lambda - \mathcal{H}_u \delta u + \lambda \frac{\mathrm{d}}{\mathrm{d}t}\delta x + \delta\lambda \dot{x}^* \right\} \mathrm{d}t + g_x \delta(X_T). \tag{25}$$

Using integration by parts, we obtain

$$\delta\mathcal{L} \approx \int_0^T \left\{ (-\mathcal{H}_x - \dot{\lambda})\delta x - \mathcal{H}_\lambda \delta\lambda + (-\mathcal{H}_u + \dot{x}^*)\delta\lambda + \frac{\mathrm{d}}{\mathrm{d}t}(\lambda\delta x) \right\} \mathrm{d}t + g_x \delta(X_T). \tag{26}$$

Let $\delta\mathcal{L} = 0$, we see that $(x^*(t), \lambda(t), u^*(t))$ satisfies the following *necessary optimality conditions* for the Hamiltonian system with optimal control:

$$x^*(0) = x_0, \qquad\qquad\qquad\qquad \text{Initial value} \tag{27}$$

$$\dot{x}^*(t) = \mathcal{H}_\lambda(x^*(t), \lambda(t), u^*(t)), \qquad \text{System dynamics} \tag{28}$$

$$\dot{\lambda}^*(t) = -\mathcal{H}_x(x^*(t), \lambda(t), u^*(t)), \quad \text{Adjoint/costate equation} \tag{29}$$

$$\lambda(T) = -g_x(x^*(T)), \qquad\qquad\qquad \text{Adjoint terminal value} \tag{30}$$

$$\mathcal{H}_u(x^*(t), \lambda(t), u^*(t)) = 0, \qquad\quad \text{Extremal} \tag{31}$$

where Eq. (28) and Eq. (29) are also known as Hamilton's canonical equations.

**Theorem B.1.** *Pontryagin's Maximum Principle (PMP) (Evans, 2024). If the $u^*$ is the optimal solution to the optimal control problem Eq. (19), then there exists a function $\lambda$ solution of the costate/adjoint equation for which*

$$u^* = \arg\max_{u \in \mathcal{U}} \mathcal{H}(x, \lambda, u), \ 0 \le t \le T. \tag{32}$$

This result implies that the Hamiltonian $\mathcal{H}$ is maximized with respect to the optimal control $u^*$ at each time $t$.

## B.3. Hamilton-Jacobi-Bellman Equation

Here, we show how to derive the expression for the optimal control. Let $p_t \in C^{2,1}(\mathbb{R}^d \times [0, T], \mathbb{R})$ be the density, then the controlled forward Fokker-Planck Kolmogorov (FPK) equation of the controlled SDE in Eq. (18) reads

$$\frac{\partial p_t}{\partial t} + \boldsymbol{\nabla} \cdot (p_t v) = \langle D(t), \nabla^2(p_t) \rangle, \tag{33}$$

where $v(x, t) := (a + \sigma u)(x, t)$ represents the controlled system dynamics and $D(t) := \frac{1}{2}\sigma^\top(t)\sigma(t)$.

Considering the optimization objective Eq. (20) with respect to the new constraint Eq. (33), we define formulate the following Lagrangian

$$\mathcal{L}(p, x, \psi) = \int_0^T \int_{\mathbb{R}^d} \left\{ h(x, u, t)p_t(x) - \psi(x, t)\Big( \underbrace{\frac{\partial p_t(x)}{\partial t}}_{\text{term (1)}} + \underbrace{\boldsymbol{\nabla} \cdot (vp_t)}_{\text{term (2)}} - \underbrace{\langle D(t), \nabla^2(p_t) \rangle}_{\text{term (3)}} \Big) \right\} \mathrm{d}x\mathrm{d}t$$

$$+ \int_{\mathbb{R}^d} g(x)p_T(x)dx, \tag{34}$$

where $\psi(x, t)$ serves as the Lagrangian multiplier.

We now apply integration by parts to term (1) with respect to $t$ and integration by parts to term (2) with respect to $x$, respectively. Then, the two-fold integration by parts are performed in term (3), and we have

$$\int_{\mathbb{R}^d} \langle D(t), \nabla^2(p_t)\rangle \psi \mathrm{d}x = \int_{\mathbb{R}^d} \langle D(t), \nabla^2(\psi)\rangle p \mathrm{d}x, \tag{35}$$

where we assume the functions have compact support such that their respective products terms vanish at both ends. Then Eq. (34) becomes

$$\mathcal{L}(p, x, \psi) = \int_0^T \int_{\mathbb{R}^d} p_t(x)\Big\{ h(x, u, t) + \frac{\partial \psi}{\partial t} + \langle \nabla\psi, v(x, t)\rangle + \langle D(t), \nabla^2(\psi)\rangle \Big\} \mathrm{d}x \mathrm{d}t$$

$$- \int_{\mathbb{R}^d} \psi(x, T) p_T(x)\mathrm{d}x + \int_{\mathbb{R}^d} \psi(x, 0) p_0(x)\mathrm{d}x. \tag{36}$$

Let the density $p_t$ be fixed, then according to the verification theorem (Zhou et al., 1997; Gozzi & Russo, 2006), we can easily attain the optimal control that minimizes Eq. 36 with respect to $u$:

$$u^* = -\sigma^\top(t)\nabla_x \psi(x, t). \tag{37}$$

Plug Eq. (37) into Eq. (36) and let the new integral equal 0, we have

$$\mathcal{L}(p, x, \psi) = \int_0^T \int_{\mathbb{R}^d} p(x, t)\Big\{ \frac{\partial \psi}{\partial t} + \frac{1}{2}\big\|\sigma^\top(t)\nabla_x \psi(x, t)\big\|_2^2 + \langle \nabla\psi, v(x, t)\rangle + \langle D(t), \nabla^2(\psi)\rangle \Big\} \mathrm{d}x \mathrm{d}t$$

$$- \int_{\mathbb{R}^d} \psi(x, T) p_T(x)\mathrm{d}x + \int_{\mathbb{R}^d} \psi(x, 0) p_0(x)\mathrm{d}x. \tag{38}$$

And the associated minimizer $(x^*(t), \lambda(t), u^*(t))$ satisfies the following optimality conditions:

$$\frac{\partial}{\partial p}\mathcal{L}(p, x, \psi) = 0 \tag{39}$$

$$\frac{\partial}{\partial p_T}\mathcal{L}(p, x, \psi) = 0. \tag{40}$$

As a result, we obtain the following partial differential equation (PDE) whose solution is the potential function $\psi$:

$$\frac{\partial \psi}{\partial t} + \frac{1}{2}\big\|\sigma^\top(t)\nabla\psi\big\|_2^2 + \langle \nabla\psi, v(x, t)\rangle = -\langle D(t), \nabla^2(\psi)\rangle, \tag{41}$$

$$\text{with the terminal condition: } \psi(x, T) = g(x), \tag{42}$$

where $\frac{1}{2}\big\|\sigma^\top(t)\nabla\psi\big\|_2^2 + \langle \nabla\psi, v(x, t)\rangle$ is the Hamiltonian with $\nabla\psi$ being the momentum. And Eq. (41) is the cerebrated Hamilton-Jacobi-Bellman (HJB) equation with the value function $\psi = \inf_u \mathcal{J}$ being the unique viscosity solution (Zhou et al., 1997; Gozzi & Russo, 2006; Yong & Zhou, 2012; Evans, 2022).

Further, Eq. (41) can be linearized by using the Hopf-Cole transformation (Evans, 2022). To this end, we let $\psi(x, t) = \log \widetilde{p}(x)$ to have:

$$\widetilde{p} = \exp(\psi) \tag{43}$$

$$\nabla\widetilde{p} = \exp(\psi)\nabla\psi \tag{44}$$

We also have

$$\langle D(t), \nabla^2(\widetilde{p})\rangle = \sum_{i,j=1}^d (D(t))_{i,j} \frac{\partial^2}{\partial x_i \partial x_j} \exp(\psi)$$

$$= \exp(\psi)\Bigg\{ \sum_{i,j=1}^d (D(t))_{i,j}\left( \frac{\partial^2 \psi}{\partial x_i \partial x_j} + \frac{\partial \psi}{\partial x_i}\frac{\partial \psi}{\partial x_j} \right) \Bigg\}$$

$$= \exp(\psi)\Bigg\{ \langle D(t), \nabla^2(\psi)\rangle + \frac{1}{2}\big\|\sigma^\top(t)\nabla\psi\big\|_2^2 \Bigg\}. \tag{45}$$

Combining Eq. (41), Eq. (44), and Eq. (45), we have

$$\frac{\partial \widetilde{p}}{\partial t} = \exp(\psi)\frac{\partial \psi}{\partial t}$$

$$= -\exp(\psi)\left\{ \frac{1}{2}\left\|\sigma^\top(t)\nabla_x\psi\right\|_2^2 + \langle\nabla\psi, v(x,t)\rangle + \langle D(t), \nabla^2(\psi)\rangle \right\}$$

$$= -\langle D(t), \nabla^2(\widetilde{p})\rangle - \langle\nabla\widetilde{p}, v(x,t)\rangle, \tag{46}$$

which in fact is the backward Fokker-Planck Kolmogorov equation, which suggests that $p_t$ is a reverse-time density. Then, the optimal control signal amounts to

$$u^* = -\sigma^\top(t)\nabla\log\widetilde{p}(x). \tag{47}$$

Consequently, according to Eq. 18, we have the following controlled reverse-time SDE:

$$dX_t = [f(X_t, t) - \sigma^2(t)\nabla\log\widetilde{p}_t(X,t)]dt + \sigma(t)d\overline{W}_t. \tag{48}$$

Recall the coupled forward-backward SDE system (Anderson, 1982; Song et al., 2020), then the above reverse-time SDE has the following forward-time counterpart:

$$dX_t = f(X_t, t)dt + \sigma(t)dW_t. \tag{49}$$

Further, we consider an overdamped Langevin dynamics system by letting $f(X_t, t) := -\nabla_x\log\hat{p}_t(x)$, where $\hat{p}_t$ is the forward-time density. Consequently, this enables us to control the sampling process in the forward (noise-to-data) sampling process.

### B.4. Path Sampling via Stochastic Optimal Control

Let $\mathbb{P} \in C^1(\mathbb{R}^d \times [0,T]; \mathbb{R}^d)$ be the base path measure and $\mathbb{P}^u \in C^1(\mathbb{R}^d \times [0,T]; \mathbb{R}^d)$ the associated path measure rendered by the optimal control $u \in C^1(\mathbb{R}^d \times [0,T]; \mathbb{R}^d)$. We have the following Radon-Nikodym derivative attained by Girsanov theorem (Liptser & Shiryaev, 2013):

$$\frac{d\mathbb{P}^u}{d\mathbb{P}} = \exp\left\{ \int_0^T u^\top(x,t)dW_t - \int_0^T \frac{1}{2}\|u(x,t)\|_2^2 dt \right\} \tag{50}$$

$$\frac{d\mathbb{P}}{d\mathbb{P}^u} = \exp\left\{ -\int_0^T u^\top(x,t)dW_t - \int_0^T \frac{1}{2}\|u(x,t)\|_2^2 dt \right\}, \tag{51}$$

where $u$ satisfies the Novikov's condition: $\mathbb{E}\left[ \exp\left(\frac{1}{2}\int_0^T u^2 dt\right) \right] < \infty$.

Noting that $dX_t = vdt + \sigma_t dW_t$, $dX_t^u = vdt + \sigma_t(dW_t + udt)$, and $\mathbb{E}\left[ \int_0^T u^\top(x,t)dW_t \right] = 0$, then the KL divergence between the two path measures amounts to

$$D_{\mathrm{KL}}(\mathbb{P}^u\|\mathbb{P}) = -\mathbb{E}_\mathbb{P}\left[ \log\frac{d\mathbb{P}^u}{d\mathbb{P}}(X) \right] = \mathbb{E}_\mathbb{P}\left[ \int_0^T \frac{1}{2}\|u(X,t)\|_2^2 dt \Big| X_0 = x \right]. \tag{52}$$

This result suggests that finding the optimal control enables us sample the target distribution. Furthermore, to reduce to the sampling variances, recalling the cost functional defined in Eq. 19, one can adopt the importance sampling scheme through sampling $N$ paths from the path measure $P^u$ and compute the average given by

$$\frac{1}{N}\sum_{i=1}^N \mathcal{J}(X^{i,u})\mathcal{W}(X^{i,u}), \tag{53}$$

where the importance weights $\mathcal{W}(X^{i,u})$ given by Eq. (51) are

$$\mathcal{W}(X^{i,u}) = \exp\left\{ -\int_0^T u^\top(x,t)dW_t - \int_0^T \frac{1}{2}\|u(x,t)\|_2^2 dt \right\}. \tag{54}$$

### B.5. Feynman-Kac Formula

The Feynman-Kac Formula is very powerful tool to solve parabolic PDEs.

**Theorem B.2.** *(Feynman-Kac formula ([Karatzas & Shreve, 2014](#))). Let $x \in \mathbb{R}^d$ be the spatial variable and $t \in [0, T]$ the temporal variable.*

$$\frac{\partial}{\partial t}\rho(x,t) + \bar{\mu}(x,t)\frac{\partial}{\partial x}\rho(x,t) + \frac{1}{2}\bar{\sigma}^2(x,t)\frac{\partial^2}{\partial x^2}\rho(x,t) - A(x,t)\rho(x,t) + c(x,t) = 0, \tag{55}$$

$$\text{subject to the terminal condition: } \rho(x,T) = \psi(x), \tag{56}$$

*where $\rho \in C^{2,1}(\mathbb{R}^d \times [0,T]; \mathbb{R})$, $\bar{\mu} \in C^1(\mathbb{R}^d \times [0,T]; \mathbb{R}^d)$, $\bar{\sigma} \in C^1(\mathbb{R}^d \times [0,T]; \mathbb{R}^{d \times d})$, $c \in C^1(\mathbb{R}^d \times [0,T]; \mathbb{R}^d)$, $A \in C^1(\mathbb{R}^d \times [0,T]; \mathbb{R}^d)$, and $\psi \in C^1(\mathbb{R}^d; \mathbb{R}^d)$ are known functions. Let $W_t$ is a Brownian motion under path measure $P$ and $X$ solves the following SDE:*

$$dX_t = \widetilde{\mu}(X,t) + \widetilde{\sigma}(X,t)dW_t. \tag{57}$$

*Then $\rho(x,t)$ can be represented by the Feynman-Kac formula as follow:*

$$\rho(x,t) = \mathbb{E}_P\left[\exp\left\{-\int_t^T A(X_s,s)\mathrm{d}s\right\}\psi(X_T) + \int_t^T \exp\left\{-\int_t^\tau A(X_s,s)\mathrm{d}s\right\}c(X_\tau,\tau)\mathrm{d}\tau \Big| X_t = x\right] \tag{58}$$

Now we use the Feynman-Kac formula to compute the marginal distribution. To this end, we first rewrite the forward Fokker-Planck equation as follows:

$$\frac{\partial}{\partial t}p(x,t) = -\boldsymbol{\nabla} \cdot (\mu p) - \langle D(t), \nabla^2(p)\rangle$$

$$= -(\boldsymbol{\nabla} \cdot \mu)p - \mu\nabla p - \langle D(t), \nabla^2(p)\rangle, \tag{59}$$

and let its coefficients match their counterparts in Eq. (55) and Eq. (56) as follows:

$$p \longrightarrow \rho \tag{60a}$$
$$\mu \longrightarrow \widetilde{\mu} \tag{60b}$$
$$\sigma \longrightarrow \widetilde{\sigma} \tag{60c}$$
$$\langle D(t), \nabla^2(p)\rangle \longrightarrow \frac{1}{2}\sigma^2\frac{\partial^2}{\partial x^2}\rho \tag{60d}$$
$$-\boldsymbol{\nabla} \cdot \mu \longrightarrow A \tag{60e}$$
$$0 \longrightarrow c \tag{60f}$$
$$g(x) \longrightarrow \psi(x). \tag{60g}$$

Therefore, according to Eq. (58), we obtain the following expressions for the marginal distribution:

$$p(x,t) = \mathbb{E}_{\mathbb{P}}\left[\exp\left\{\int_t^T \boldsymbol{\nabla} \cdot \mu(X_s,s)\mathrm{d}s\right\}g(X_T)\Big|X_t = x\right] \tag{61}$$

$$p(x,t) = \mathbb{E}_{\mathbb{P}}\left[\exp\left\{-\int_0^t \boldsymbol{\nabla} \cdot \mu(X_s,s)\mathrm{d}s\right\}g(X_0)\Big|X_0 = x\right]. \tag{62}$$

Combining Eq. (61) with Eq. 52, we use Jensen's inequality to obtain the following ELBO:

$$\log p(x,t) \geq \mathbb{E}_{\mathbb{P}}\left[-\int_0^t \left\{\boldsymbol{\nabla} \cdot \mu(X_s,s)\mathrm{d}s + \log g(X_0)\Big|X_0 = x\right\} - \mathbb{E}_{\mathbb{P}}\left[\log \frac{\mathrm{d}\mathbb{P}^u}{\mathrm{d}\mathbb{P}}(X)\right]\right.$$

$$\geq \mathbb{E}_{\mathbb{P}}\left[-\int_0^t \left\{\boldsymbol{\nabla} \cdot \mu(X_s,s) + \frac{1}{2}\|u(X,s)\|_2^2\right\}\mathrm{d}s + \log g(X_0)\Big|X_0 = x\right] \tag{63}$$

### B.6. Connection to Flow Matching

Since we have the intermediate sample $X_t = \frac{t}{T}X_T + (1 - \frac{t}{T})X_0$, we can directly computed the predicted terminal samples at time $T$ using the predicted $\hat{X}_t$ at time $t$ without iterative function evaluations:

$$\hat{X}_T \approx \left(\hat{X}_t - (1 - \frac{t}{T})X_0\right)/(\frac{t}{T}). \tag{64}$$

Assume the log-densities of $X_0$, $X_t$, and $X_T$ can be represented by the same function, then the terminal cost in the value function Eq. (20) is defined as $g(X_T) := -\log p(X_T)$. As a result, it suggests that minimizing the running cost at time $t$ also means minimizing the terminal cost at time $T$.

First, following the principle of least action in Lagrangian dynamics, the uncontrolled system dynamics $v(x_t, t)$ is learned by minimizing the associated kinetic energy.

The uncontrolled sampling process can be described as follow:

$$dx_t = v(x_t, t)\mathrm{d}t \tag{65}$$

Since we formulate the potential energy function $U(x_t, t) = \log p(x_t, t) = \log \mathcal{N}(x_t; \tilde{x}_t, \tilde{\sigma}_t)$, where $\tilde{x}_t = \text{TDAE}(x_t, t)$ attained by the reconstruction process of the TDAE model. Accordingly, the controlled sampling process can be cast as:

$$dx_t = u(x_t, t)dt = -\sigma^\top(t)\nabla_x \log p(x_t, t)\mathrm{d}t = -\frac{x_t - \tilde{x}_t}{\tilde{\sigma}_t^2} \tag{66}$$

The corresponding sampling scheme is

$$x_{t+1} = x_t + v(x_t, t)\mathrm{d}t \tag{67}$$
$$x_{t+1}^u = x_{t+1} + u(x_{t+1}, t+1)\mathrm{d}\tilde{t}. \tag{68}$$

## C. Rao-Blackwellization

Here we prove that the proposed sampler is, in fact, a Rao-Blackwellized trajectory sampler (Casella & Robert, 1996). We first start with the following definition:

**Definition C.1** (Sufficient statistic). A *sufficient statistic* $\mathcal{T}$ for a parameter $\Theta$ captures all the necessary information contained in the data sample $\mathcal{X}$ to estimate $\Theta$. Once $\mathcal{T}$ is known, $\mathcal{X}$ does not provide additional information to estimate $\Theta$.

To determine whether a statistic is sufficient, we can apply the following theorem.

**Theorem C.2.** *(Fisher-Neyma theorem (Lehmann & Casella, 2006)). Let probability density function of $\mathcal{X}$ be $p(x|\varphi)$, then the statistics $\mathcal{T}$ are sufficient for $\mathcal{X}$ iff $p(x|\varphi)$ are be written in the following form:*

$$p(x|\varphi) = \mathcal{F}(x)\mathcal{G}(\mathcal{T}(x); \varphi), \tag{69}$$

*where $\mathcal{F}(x)$ is a distribution independent of $\theta$ and $g(\cdot, \theta)$ captures all the dependence on $\theta$ via sufficient statistics $\mathcal{T}(x)$.*

Following the above theorem, in our context, we assume that the marginal distribution of $X_t$ is the Gaussian with unknown mean and known variance: $\mathcal{N}(X_t; m_\varphi(X_{t-1}), \sigma^2(t))$. Then the joint distribution of $N$ samples can be written and decomposed as follows:

$$p(X_t^1, X_t^2, \ldots, X_t^N|\varphi) = (2\pi)^{-N/2}\sigma^2 \exp\left(\frac{-1}{2\sigma^2}\sum_{i=1}^{N}(X_t^i - m_\varphi)^2\right)$$

$$= (2\pi)^{-N/2}\sigma^2 \exp\left(\frac{-1}{2\sigma^2}\sum_{i=1}^{N}X_t^i + \frac{m_\varphi}{\sigma^2}\sum_{i=1}^{N}X_t^i - \frac{Nm_\varphi^2}{2\sigma^2}\right)$$

$$= (2\pi)^{-N/2}\sigma^2 \underbrace{\exp\left(\frac{-1}{2\sigma^2}\sum_{i=1}^{N}X_t^i\right)}_{\mathcal{F}(x)} \underbrace{\exp\left(\frac{m_\varphi}{\sigma^2}\underbrace{\sum_{i=1}^{N}X_t^i}_{\mathcal{T}(x)} - \frac{Nm_\varphi^2}{2\sigma^2}\right)}_{\mathcal{G}(\mathcal{T}(x);\varphi)}. \tag{70}$$

The above result suggests that the trajectory sampler can be formulated as a sufficient statistic for $\varphi$. Consequently, for our task, we have: 1) the parameter to estimate $\Theta := X_t$, where $X_t$ is the intermediate sample predicted (or its mean) at time $t$; 2) the data sample $\mathcal{X} := X_0$, where $X_0$ is the initial noise (as well as the observation); 3) the base unbiased sampler representing the system dynamics estimated according to the principle of least action, $\mathcal{S}(X_0; \mu, t) := X_0 + \int_0^t \mu(X_s, s)\mathrm{d}s = X_{t-1} + \int_{t-1}^t \mu(X_s, s)\mathrm{d}s = X_t$, where $0 < t \leq T$; 4) the sufficient statistic representing the optimal control signal according to Pontryagin's Maximum Principle, $\mathcal{T}(X_0; u, t) := X_0 + \int_0^{t-1} u^*(X_s, s)\mathrm{d}s = X_{t-1}$, where $0 < t \leq T$. As a result, we have the new sampler $\mathcal{S}^* := \mathbb{E}[\mathcal{S}|\mathcal{T}] = \mathbb{E}[\mathcal{T}(X_0; u, t) + \int_{t-1}^t \mu(\mathcal{T}(X_0; u, s), s)\mathrm{d}s]$. Then, according to the Rao-Blackwell theorem:

**Theorem C.3.** *(Rao-Blackwell theorem (Casella & Robert, 1996)). Let $\mathcal{S}$ be an unbiased estimator of some parameter $\Theta$, and $\mathcal{T}(\mathcal{X})$ the sufficient statistic for $\Theta$, then: 1) $\mathcal{S}^* = \mathbb{E}[\mathcal{S}|\mathcal{T}(\mathcal{X})]$, is an unbiased estimator for $\Theta$, and 2) $\mathbb{V}[\mathcal{S}] \geq \mathbb{V}[\mathcal{S}^*]$. The inequality is strict unless $\mathcal{S}$ is a function of $\mathcal{T}$.*

**Proof:** In the ODE/SDE sampling process, we have $p(X_t|X_{t-1}, X_0) = p(X_t|X_{t-1})$, i.e., $p(\Theta|\mathcal{T}, \mathcal{X}) = p(\Theta|\mathcal{T})$. Since $\mathcal{T}$ is a statistic of $\mathcal{X}$ and $\mathcal{S}$ is an estimator of $\Theta$, we have $\mathbb{E}[\mathcal{S}|\mathcal{T}] = \mathbb{E}[\mathcal{S}|\mathcal{T}, \Theta]$. We now apply the law of total expectation ($Z$ and $Y$ are two random variables):

$$\mathbb{E}[Z|Y] = \int zp(z|Y)\mathrm{d}z \implies \mathbb{E}[\mathbb{E}[Z|Y]] = \iint zp(z|y)\mathrm{d}zp(y)\mathrm{d}y$$
$$= \iint zp(z|y)p(y)\mathrm{d}z\mathrm{d}y$$
$$= \iint zp(z, y)\mathrm{d}z\mathrm{d}y = \int zp(z)\mathrm{d}z = \mathbb{E}[Z], \qquad (71)$$

to attain the following relationships:

$$\mathbb{E}[\mathcal{S}^*|\Theta] = \mathbb{E}[\mathbb{E}[\mathcal{S}|\mathcal{T}]|\Theta] = \mathbb{E}[\mathbb{E}[\mathcal{S}|\mathcal{T}, \Theta]|\Theta] = \mathbb{E}[\mathcal{S}|\Theta]. \qquad (72)$$

Then we apply the law of total variance to attain the following relationships:

$$\mathbb{V}[\mathcal{S}|\Theta] = \mathbb{E}[\mathbb{V}[\mathcal{S}|\mathcal{T}, \Theta]|\Theta] + \mathbb{V}[\mathbb{E}[\mathcal{S}|\mathcal{T}, \Theta]|\Theta]$$
$$= \mathbb{E}[\mathbb{V}[\mathcal{S}|\mathcal{T}, \Theta]|\Theta] + \mathbb{V}[\mathcal{S}^*|\Theta], \qquad (73)$$

where $\mathbb{E}[\mathbb{V}[\mathcal{S}|\mathcal{T}, \Theta]|\Theta] \geq 0$, therefore $\mathbb{V}[\mathcal{S}|\Theta] \geq \mathbb{V}[\mathcal{S}^*|\Theta]$.

The results in Eq. (72) and Eq. (73) suggest that the new sampler has the same expectation as the base sampler but with smaller variance (mean squared error), which is also verified by the experimental results.

### C.1. Overall Theoretical Framework

Fig. 3 visualizes the overall theoretical framework of the proposed method in this paper.

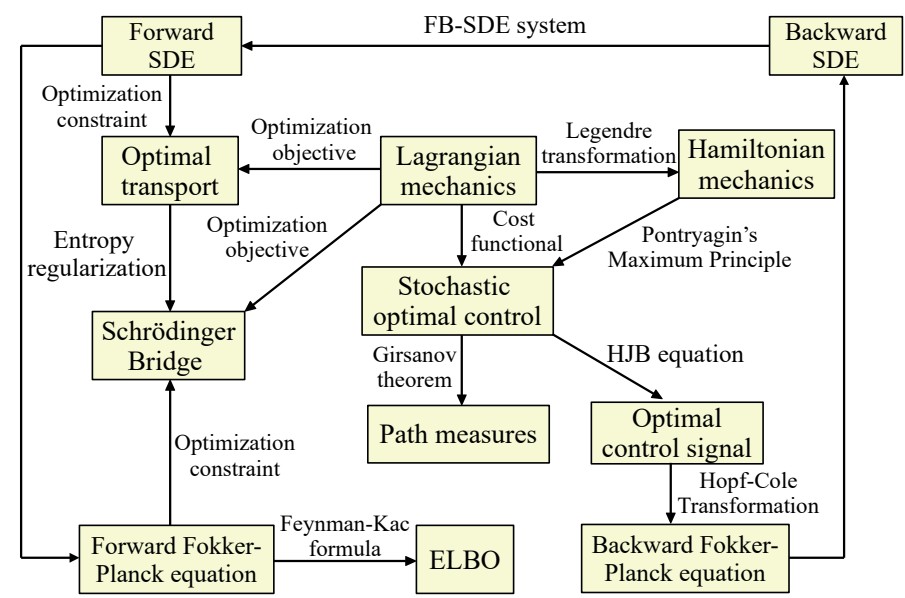

Figure 3: The overall theoretical framework.

## D. Resampling Trick

The proposed resampling process introduced in Sec. 3.5 is illustrated in Fig. 4.

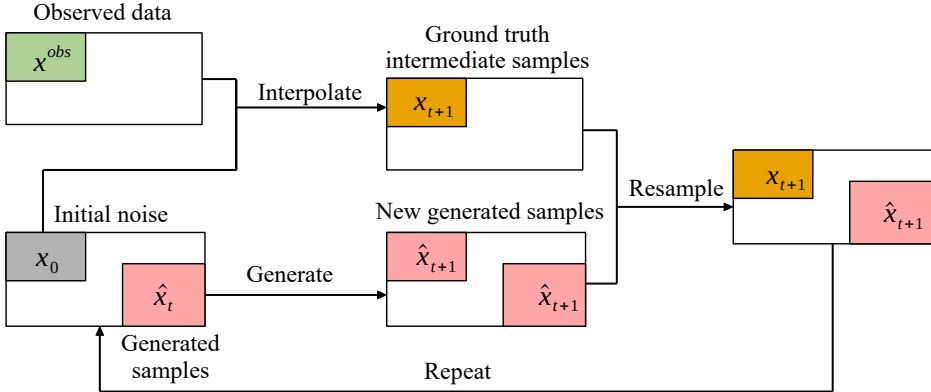

Figure 4: The proposed resampling process for inference.

## E. Experimental Environment

For the hardware environment of the experiments, we use a single NVIDIA A100-PCIE-40GB GPU and an Intel(R) Xeon(R) Gold-6248R-3.00GHz CPU. For the software environment, the Python version is 3.9.7, the CUDA version 11.7, and the Pytorch version is 2.0.1.

## F. Additional Experimental Results

### F.1. Ablation study on simulation steps.

Table 6: Test imputation results on PM 2.5 with different simulation steps (5 trials).

| Method | 5 steps | | 10 steps | | 15 steps | | 20 steps | |
|--------|---------|---|----------|---|----------|---|----------|---|
| | RMSE | MAE | RMSE | MAE | RMSE | MAE | RMSE | MAE |
| CSDI | $34.21 \pm 0.16$ | $14.85 \pm 0.01$ | $29.43 \pm 0.46$ | $12.48 \pm 0.08$ | $22.40 \pm 0.16$ | $10.78 \pm 0.04$ | $19.22 \pm 0.13$ | $9.91 \pm 0.02$ |
| CLWF | $\mathbf{18.29 \pm 0.002}$ | $\mathbf{9.78 \pm 0.004}$ | $\mathbf{18.28 \pm 0.003}$ | $\mathbf{9.77 \pm 0.005}$ | $\mathbf{18.26 \pm 0.006}$ | $\mathbf{9.76 \pm 0.004}$ | $\mathbf{18.21 \pm 0.002}$ | $\mathbf{9.72 \pm 0.004}$ |

### F.2. Ablation study on Rao-Blackwellization.

Table 7: Test imputation results on PM 2.5, PhysioNet 0.1, and PhysioNet 0.5 (5 trials).

| Method | PM 2.5 | | PhysioNet 0.1 | | PhysioNet 0.5 | |
|--------|--------|---|---------------|---|---------------|---|
| | RMSE | MAE | RMSE | MAE | RMSE | MAE |
| CLWF (no RB) | $18.27 \pm 0.01$ | $9.76 \pm 0.01$ | $0.4802 \pm 1e\text{-}4$ | $\mathbf{0.2221 \pm 0e\text{-}4}$ | $0.6476 \pm 0e\text{-}4$ | $\mathbf{0.2991 \pm 0e\text{-}4}$ |
| CLWF (with RB) | $\mathbf{18.08 \pm 0.02}$ | $\mathbf{9.71 \pm 0.00}$ | $\mathbf{0.4785 \pm 1e\text{-}4}$ | $0.2250 \pm 1e\text{-}4$ | $\mathbf{0.6466 \pm 0e\text{-}4}$ | $0.3003 \pm 0e\text{-}4$ |

Table 8: Test imputation results on synthetic data (5-trials, values are multiplied by $10^2$).

| Method | Synthetic 0.4 | | Synthetic 0.6 | | Synthetic 0.8 | |
|--------|---------------|---|---------------|---|---------------|---|
| | RMSE | MAE | RMSE | MAE | RMSE | MAE |
| CLWF (no RB) | $22.91 \pm 0.49$ | $15.28 \pm 0.21$ | $25.65 \pm 0.31$ | $15.54 \pm 0.22$ | $27.41 \pm 0.27$ | $15.91 \pm 0.23$ |
| CLWF (with RB) | $\mathbf{22.72 \pm 0.48}$ | $\mathbf{13.23 \pm 0.42}$ | $\mathbf{25.44 \pm 0.30}$ | $\mathbf{15.28 \pm 0.17}$ | $\mathbf{27.32 \pm 0.27}$ | $\mathbf{15.79 \pm 0.23}$ |

### F.3. Ablation study on Resampling.

Table 9: Test imputation results on ETT-h1(5-trial averages). The best are in bold and the second best are underlined.

| Method | ETT-h1 0.25 | | ETT-h1 0.375 | | ETT-h1 0.5 | |
|--------|-------------|---|--------------|---|------------|---|
| | RMSE | MAE | RMSE | MAE | RMSE | MAE |
| Base | $0.1999 \pm 8e\text{-}4$ | $0.1317 \pm 3e\text{-}4$ | $0.2191 \pm 2e\text{-}4$ | $0.1422 \pm 4e\text{-}4$ | $\underline{0.2906 \pm 1e\text{-}3}$ | $\underline{0.1882 \pm 2e\text{-}4}$ |
| RB | $\underline{0.1970 \pm 6e\text{-}4}$ | $0.1266 \pm 2e\text{-}4$ | $0.2185 \pm 4e\text{-}4$ | $0.1424 \pm 2e\text{-}4$ | $\mathbf{0.2891 \pm 2e\text{-}4}$ | $\mathbf{0.1845 \pm 2e\text{-}4}$ |
| Resampling | $0.1976 \pm 8e\text{-}4$ | $\underline{0.1257 \pm 3e\text{-}4}$ | $\underline{0.2165 \pm 1e\text{-}3}$ | $\underline{0.1366 \pm 1e\text{-}4}$ | $0.2964 \pm 1e\text{-}3$ | $0.1988 \pm 4e\text{-}4$ |
| Resampling + RB | $\mathbf{0.1968 \pm 8e\text{-}4}$ | $\mathbf{0.1253 \pm 3e\text{-}4}$ | $\mathbf{0.2157 \pm 6e\text{-}4}$ | $\mathbf{0.1363 \pm 3e\text{-}4}$ | $0.2921 \pm 9e\text{-}4$ | $0.1926 \pm 3e\text{-}4$ |

### F.4. Time Efficiency

We report the statistics regarding the time efficiency of our method here.

Table 10: Inference time costs on ETT-h1 0.5.

| | Base | RB | Resampling | Resampling + RB |
|--------|------|-----|------------|-----------------|
| s/iteration | 3.26 | 6.36 | 3.28 | 6.38 |