# OpenReview forum: "Conditional Lagrangian Wasserstein Flow for Time Series Imputation"
_ICML.cc/2025/Conference — Submitted to ICML 2025_

### Official Review · Reviewer_CsBE · 2025-03-02

**Overall Recommendation:** 3

**Summary:**

The paper proposes a time-series imputation method based on optimal transport flow matching. To improve point-estimation of the imputations, the paper suggests learning an additional denoising autoencoder, which when used during sampling reduces imputation variance. Evaluation with common point-estimation metrics shows favorable performance of the new method.

**Claims And Evidence:**

1.  Paragraph 3 in the introduction claims that "we treat the multivariate time series imputation task as a conditional optimal transport problem, whereby the random noise is the source distribution, the missing data is the target distribution, and the observed data is the conditional information." This suggests that the paper aims to learn the conditional _distribution_ of the missing data.
	* However, the addition of the drift term in $v_t^{\phi}$ in the sampling procedure (Algorithm 2) pushes the samples towards the conditional mean of the missing data distribution. As a result, the approach may potentially ignore the uncertainty, as is also highlighted at the end of Appendix C.
	* It is therefore important to highlight that the addition of this drift term, may be used to trade-off imputation uncertainty for lower-variance imputations (that may be close to the conditional mean of the missing distribution).
	* While singular imputation methods that aim to impute data with samples from the conditional mean (alternatively, mode or median) have important applications, it is also important to highlight the known caveats of singular imputation as compared to multiple imputation that aims to sample the distribution of missing values, see e.g. Section 2.1.2 of [1]. As such, I believe that the paper could benefit its readers by more clearly stating the imputation goals, potentially highlight caveats of singular imputation if that is the main goal, and elaborating that "reduc[ing] the sampling variances" can in fact reduce the probabilisitic method to a singular imputation method.

2. Please also provide a reference or a proof to the statement in second column of Line 169. Namely, please elaborate how equation 13 minimises the kinetic energy in equation 7.

References:

[1] van Buuren (2018). Flexible Imputation of Missing Data.

**Essential References Not Discussed:**

No additional references are needed, but the paper should highlight where the copied baseline results come from.  E.g. some of them seem to be copied from [1], but other may be from elsewhere.

References:

[1] Chen et al (2023). Provably Convergent Schrödinger Bridge with Applications to Probabilistic Time Series Imputation.

**Experimental Designs Or Analyses:**

The evaluations in Section 4 seem ok for point-wise estimation. However, the goals in the introduction must clearly state whether the paper aims to solve singular or multiple imputation. For example, if the goal is multiple imputation, then the ablation in Table 3 does not make much sense, since CSDI inherently samples the distribution of the missing data and as such any increase in the point-wise metrics may be attributed to the inherent uncertainty of the true missing data distribution.


Some of the baseline results in table 1 are copied from [1], which is ok, but should be acknowledged where appropriate! Highlighting whether there may be any differences in the evaluation settings would also be helpful.

References:

[1] Chen et al (2023). Provably Convergent Schrödinger Bridge with Applications to Probabilistic Time Series Imputation.

**Methods And Evaluation Criteria:**

The method proposed in Section 3 is sound to the best of my understanding, and provides a way to trade-off sampling the correct missing data distribution for samples that are closer to the conditional mean by using the learnt drift term $v_t^{\phi}$. However, it is important to assess this trade-off empirically using probabilistic metrics:

* The current evaluations focused only on point-wise metrics, such as MAE and RMSE.
	* The paper should evaluate the proposed method using some probabilistic metrics, such as CRPS, MMD, Wasserstein distance, and so on [1,2,3]. I would expect that the Rao-Blackwellization ablations in Table 5 would reveal that Rao-Blackwellization reduces the performance on these probabilistic metrics, since it encourage the sampler to be close to the conditional mean of the missing values, rather than _sampling_ the true distribution of missing values.

References:

[1] Muzellec et al (2020). Missing Data Imputation using Optimal Transport

[2] Jolicoeur-Martineau et al (2024). Generating and Imputing Tabular Data via Diffusion and Flow-based Gradient-Boosted Trees

[3] van Buuren (2018). Flexible Imputation of Missing Data

**Other Comments Or Suggestions:**

Typos:
* Line 105: "equatio" -> "equation"
* Line 137: I believe the $\mathbb{R}$ should be $\{0, 1\}$ as the mask $M$ should be binary.
* Line 200: "learnning" -> "learning"
* Line 201: "introduce" -> "introduced"
* Line 266: "flowing" -> "flow"
* Line 913: "Neyma" -> "Neyman"

**Other Strengths And Weaknesses:**

The paper is fairly well-written and presents a sufficiently novel method. The results on point-estimation are fairly good and will be of interest to some practitioners interested in missing data imputation. However, it is important to improve the rigour of the problem setting and the goals of the approach (e.g. does the paper aim to generate singular imputation, sample multiple imputation, or provide a controllable way between the two?).

**Questions For Authors:**

No further questions, please see the questions/concerns above. To me, the most important aspect to address is the problem setting and highlighting the potential caveats of Rao-Blackwellizing the imputations.

**Relation To Broader Scientific Literature:**

The paper builds heavily on CSDI [1] by using a similar model specification, architecture, and evaluation. The authors replace score matching with a combination of flow-matching and the Rao-Blackwellization term, which IMO is sufficiently novel and shows improved point-estimation performance. However, as the paper drops the probabilistic metric (CRPS) from [1], I assume that it may have deteriorated, and thus this highlighting performance drop could be useful to the readers of the paper.


References:

[1] Tashiro et al (2021). CSDI: Conditional Score-based Diffusion Models for Probabilistic Time Series Imputation

**Theoretical Claims:**

The theoretical results and proofs in the Appendix seem correct, but as discussed above the potential caveats of "reducing sampling variance" (via Rao-Blackwellization) need to be discussed and evaluated.

---

> ### Author Rebuttal · Authors · 2025-03-31
>
> Thank you very much for the insigtful comments (length limit, the response is concise).
> ### **Claims And Evidence**
> 1. > - However, the addition of the drift term...
> #### **Response**: The proposed theoretical framework can be used to analyze the learning process of the conditional distribution. However, in realization we find that estimating the conditional mean of the distribution can achieve better empirical performance in terms of RMSE and MAE.The Rao-Blackwellization only focus on reducing the variance of the sampler; which may disregard the uncertainty. Nevertheless, this can also be beneficial for reducing the sampling time for the base unbiased sampler. Because after the Rao-Blackwellization we can use the same number of Monte Carlo path samples to estimate the sample means more accurately. Please refer to the experimental results shown in Appendix F.2.
>  > -  ... elaborating that "reduc[ing] the sampling variances" can in fact reduce the probabilisitic method to a singular imputation method.
> #### **Response**: Thanks for the suggestion. The realization of the proposed method is more suitable for singular imputation. We will add following content in the main paper: “If we aim for singular imputation, i.e., estimating the conditional means of the missing data, the Rao-Blackwellization can be used to reduce the sampling variances. However, please note that  the sampling diversity may decrease.”
> > 2. Please also provide a reference or a proof to the statement in second column of Line 169....
> #### **Response**: According to the definition of the dynamical OT/Schrodinger bridge problem given by Eq (5), the infimum of which is in fact the Wasserstein distance between the initial and target distributions. And the velocity field $\mu_t$ satisfies the Fokke-Planck equation, Eq (2). The kinetic energy functional endowed by Eq (7) is minimized when the transport path is the geodesic in the Wasserstein space.Since we have the velocity field $dx/dt = \mu_t(x) = \frac{X_T-X_0}{T-0}$, and the optimal solution satisfies $\Pi_t (x) = (1-t)x + tT(x)$; where $T$ is the optimal pushforward map. Then the velocity field  $\mu_t(x) = dx/dt = T(X)-X$ and its norm: $||\mu_t(x)||^2 =||T(X)-X||^2$. The corresponding kinetic energy reads $K = \int^T_0 \int_R ||T(x) -x||^2dp(x_t)dt$, where the intermediate sample $p(x_t)$ is rendered by interpolating the initial and target distributions using Eq (7). Thus, $K$ evaluates the Wasserstein distance between the initial and target distributions. According to the Cauchy-Schwarz inequality, any other path will have greater kinetic energy.We will include this proof in the appendix. Reference:
> ##### Neklyudov, eat al. A computational framework for solving wasserstein lagrangian flows. ICML, 2024.
> ### **Methods And Evaluation Criteria**
> > The paper should evaluate the proposed method using some probabilistic metrics...
> #### **Response**: Thanks. We show the performance comparison between the base sampler and the Rao-Blackwellized one on ETTh1.
>
> --- | MMD | Wasserstein| CRPS|
> --- | --- | --- |---- |
> Base| 0.13| 52.42 | 0.19|
> RB   | 0.12| 48.70 | 0.19|
>
> This shows that reducing variances may be beneficial for the performance in our case. We will highlight that RB may degrade the model’s performance on probabilistic metrics in the paper.
>
> ### **Experimental Designs Or Analyses**
> >However, the goals in the introduction must clearly state whether the paper aims to solve singular or multiple imputation.
> #### **Response**:  As we stated in Sec 2.1. The theoretical is based on the SDE formulation of the OT problem as it provides a general framework. However, in practice, we opt for the ODE sampler to achieve the best empirical results for singular imputation. We will highlight the above explanation in the paper.
> >Some of the baseline results in table 1 are copied from [1]
> #### **Response**: Thanks for the suggestion. The results will be explicitly acknowledged in the experiment part of the main text as follows: Some of results of the baselines shown in Table 1 are reported from [1].
> ### **Relation To Broader Scientific Literature**
> #### **Response**: Indeed, reducing the variance of samples may damage the model’s probabilistic modeling performance. The reason we opt for non-probabilistic metrics to evaluate the model because the designed base sampler is based on ODE which attains better RMSEs and MAEs compared to existing methods. Moreover, CRPS may not be a suitable evaluation metric in our case. [1] and [2] suggest that CRPS could lead to an incorrect assessment of overall model performance by overlooking the model's performance on each dimension of data.
> ##### [1] Alireza, et al. Random noise vs. state-of-the-art probabilistic forecasting methods: A case study on CRPS-Sum discrimination ability. Applied Sciences, 2022.
> ##### [2] Marin, et al. Modeling temporal data as continuous functions with stochastic process diffusion. ICML, 2023.
>
> #### **Typos**
> Thanks. We will revise the paper.

---

> > ### Comment · Reviewer_CsBE · 2025-04-05
> >
> > Thanks for the response. I am mostly happy with the response and keep my original recommendation.

---

> > > ### Author Response · Authors · 2025-04-05
> > >
> > > Thank you for your feedback. We are glad that you like the reponse.

---

### Official Review · Reviewer_n7gf · 2025-03-07

**Overall Recommendation:** 1

**Summary:**

In this paper, the authors proposed a novel time-series imputation approach named `Conditional Lagrangian Wasserstein Flow` (CLWF) based on the functional optimization approaches, for example, Schr \"{o}$dinger bridge, optimal transport, flow matching, etc. At first, the authors reformulated the data imputation problem as an optimization of some potential function, which can be solved by simulating some dynamical system under the principle of least action. After that, the authors designed the time-dependent denoising autoencoder (TDAE) to estimate the gradient of potential function. Based on the abovementioned contents, the authors summarized the algorithms for TDAE training and CLWF sampling. Finally, experimental results are conducted on related datasets to demonstrate the efficacy of the proposed approach.

## update after rebuttal

I have carefully read the author's reply and I think my concerns have not been addressed. This article lacks clear motivation, the understanding of the concept of "diff" is unclear, and there might be problems in the code implementation (averaging of the inference results, not calculating the mask matrix).

**Claims And Evidence:**

1. In page one, the authors mentioned that diffusion models `are limited to slow convergence or large computational costs.` To the reviewer's knowledge, the acceleration of diffusion models has been attempted throughout previous works like DPM-Solver [1] and Diffusion Exponential Integrator Sampler [2]. The claim was made without considering these progresses.

2. In page 1, right column, `In our method, we treat the multivariate time series imputation task as a conditional optimal transport problem, whereby the random noise is the source distribution, the missing data is the target distribution, and the observed data is the conditional information.` It seems that the missing mechanism was ignored in this procedure, which indicates that the proposed approach may not be suitable for the missing not at random (MNAR) scenario, a familiar case in recommender systems.

3. In Eq. (5), the authors have not provided the boundary condition, to the reviewer's knowledge, the solving of optimal control or Schr$\"o$dinger bridge should include the boundary condition.

4. In Page 3, right column, the problem statement of the time-series imputation task has not been given in detail, for example, is $M_{i,j}\in\{0,1\}$?

5. In page 4, the authors stated that `Note that Eq. (12) can only allow us to generate time-dependent intermediate samples in the Euclidean space but not the Wasserstein space, which can lead to slow convergence as the sampling paths are not straightened.` Is it the reason that Eq. (12) does not satisfy the Fokker-Planck equation that delineates the conservation of density in Wasserstein space? Please give detailed proof.

6. In the supplementary material, the authors stated that `Therefore, the data generation task can also be interpreted as a stochastic optimal control (SOC) problem`. It seems that the theory that the data generation problem can be treated as an SOC problem is based on the reason that the SOC formulates an upper bound of the KL divergence between generated samples and original data samples, as supported by reference [3], this statement has not been clarified.

---

References
[1]. DPM-Solver: A Fast ODE Solver for Diffusion Probabilistic Model Sampling in Around 10 Steps, NeurIPS 2022

[2]. Fast Sampling of Diffusion Models with Exponential Integrator, ICLR 2023

[3]. Path Integral Sampler: a stochastic control approach for sampling, ICLR 2022

**Essential References Not Discussed:**

See the references listed aboved.

**Experimental Designs Or Analyses:**

1. Notably, the authors have proposed the optimal transportation based approach, but the baseline comparison has not included static optimal transportation approaches, for example references [1], [2].

2. The computational time comparison in Table 10 should include those of baseline models.

---

References
[1]. Missing Data Imputation using Optimal Transport, ICML 2020

[2]. Transformed Distribution Matching for Missing Value Imputation, ICML 2023

**Methods And Evaluation Criteria:**

1. Proposed method: Treating the data imputation task as a data generation task, which introduces the noise during the imputation procedure, may not be suitable. Specifically, reference [1] has explicitly pointed out that `Specifically, diversity is a key objective of the generation problem, which requires the generated data to vary significantly while maintaining relevance to the given context. Diffusion models being sensitive to the initial noise ($x_T$) at the generation stage helps generate diverse samples – different noise usually leads to different generated samples. Conversely, the objective of the imputation task is accuracy rather than diversity, requiring the imputed data to closely resemble the singular groundtruth values.` The proposed approach also generates missing data from noise, which seems to go against this statement. More validations need to be provided to support the proposed method. Based on this, reference [2] has proven this issue from the perspective of gradient flow, a mathematical tool based on the ODE and PDE.

2. Evaluation: Tables 1 to 5 have not conducted the paired-sample $t$-test.

3. The overall algorithm for CLWF has not been given, suppose when we have a missing dataset at hand, how can we use the CLWF *ab initio* to fill the missing data?


References:
[1]. Self-Supervision Improves Diffusion Models for Tabular Data Imputation, CIKM 2024

[2]. Rethinking the Diffusion Models for Missing Data Imputation: A Gradient Flow Perspective, NeurIPS 2024

**Other Comments Or Suggestions:**

Refer to the above contents.

**Other Strengths And Weaknesses:**

### Strengths

1. The topic is related to the ICML conferences.
2. The proposed approach seems to be interesting.

### Weaknesses

1. There is no need for introducing concepts like Schrodinger bridge in the main content, since the main content has not used this concept.
2. The detailed evaluation metric computational protocols should be included in the manuscript.

**Questions For Authors:**

Refer to the above contents.

**Relation To Broader Scientific Literature:**

See the references listed aboved.

**Theoretical Claims:**

1. Eq. (46). It seems that the optimal control policy given by (46) is the solution of some infinite optimal control problem, can we further extend this to the finite horizon problem?

2. In appendix B.6 it seems that the flow matching is based on the first order dynamical system, but the Lagrangian mechanics, which involves Hamiltonian computation, is a second order dynamic, shall we understand this from a higher order system?

3. In Section 3.4 shall we use $U_t(X_t) \propto -\log{\mathcal{N}(X_t\vert \widehat{X}_t, \sigma_p^2)}$?

---

> ### Author Rebuttal · Authors · 2025-03-29
>
> Thank you very much for the insigtful comments (due to length limit, we have to keep the response concise here).
> ### **Claims And Evidence**
>  > 1.To the reviewer's knowledge, the acceleration of diffusion models ...
> #### **Response**: Here, we mainly refer to the diffusion models for time series imputation tasks, e.g., CSDI. Indeed, DPM-Slover and Diffusion Exponential Integrator Sampler can accelerate the inference procedures for diffusion models. However, CLWF can use less model evaluation steps to achieve competitive results (please refer to the experimental results in Appendix F.1 on Page 20). We will discuss the above works in the paper.
>  >2. It seems that the missing mechanism ...
> #### **Response**:  The proposed method is able to impute unconditionnaly, thus it maybe able to deal with this problem to some extent.  However, we will investigate this issue in the future work.
> >3. In Eq. (5), the authors have not provided the boundary condition, ...
> #### **Response**:  The boundary condition is given by the definition of SDE in Eq (1) by letting the initial distribution equal $p(X_0)$ and the target distribution equal $p(X_T)$ using stochastic interpolating.
> >4. is $M_{i,j} \in0,1$?
> #### **Response**: Yes, we will define the problem more rigorously.
> >5.  Is it the reason that Eq. (12) does not satisfy the Fokker-Planck equation that...
> #### **Response**: If the Eq (12) performed in Wasserstein space via projection, then it still satisfies the Fokker-Planck equation.
> >6.  It seems that the theory that the data generation problem can be treated as an SOC...
> #### **Response**:  The KL divergence between the sampling path measure (related to the controlled SDE) and the reference path measure (related to the uncontrolled SDE) can be minimized by solving the corresponding SOC problem according to Girsanov theorem. Please also refer to Appendix B.4.
> ### **Methods And Evaluation Criteria**
> >1. ...proposed approach also generates...
> #### **Response**: If we can estimate the target distribution mean unbiasedly, then a generative methodology is suitable for imputation tasks.
> >2. We will include this in the paper.
> >3. The training procedure of CLWF in described in Algorithm 1 in Sec 3.6 on Page 5 and the inference procedure of CLWF is described in Algorithm 2 in Sec 3.6 on Page 6.
>
> ### **Theoretical Claims**
> >1. It seems that the optimal control policy given by (46)...?
> #### **Response**:  Eq. (46) in fact solves a finite control problem from $t_0$ to $t_T$. It is possible to extent this to the finite horizon problem if we can learn a control function that asymptotically stabilizes the system by satisfying the Lyapunov stability function. In fact, [1] proves that minimize the free energy functional for all probability densities satisfies the Lyapunov stability.
> ##### [1] H. Risken, The Fokker-Planck equation: Methods of solution and applications, 2nd ed., Springer-Verlag, Berlin, Heidelberg, 1989;
> >2. In appendix B.6 it seems that the flow matching is based on the first order dynamical system, ...?
> #### **Response**: The second order term is involved in the Lagrangian mechanics (the text above Appendix B.4 on Page 15). However, we adopt a sampler based on the overdamped Langevin dynamics, in which the acceleration is assumed to be negligible due to the existence of strong friction. $md^2x/dt^2 = -\lambda dx/dt - \nabla U(x) + \sqrt{2M}dt = 0$, therefore, the second order system is reduced to a first order system: $\lambda dx/dt = - \nabla U(x) + \sqrt{2M}dt$.
> >3. In Section 3.4 shall we use $U_t(X_t) \propto −\log ⁡N(X_t|\hat{X}_t,\sigma_p^2)$?
> #### **Response**: Thanks for the suggestion. This may be more rigorous.
> ### **Experimental Designs Or Analyses**
> >1.
> Method| ETT-h1 |0.25|  ETT-h1 |0.375 |ETT-h1 |0.5 |
> --- | --- | --- |---- |---- | --- |---- |
> --- | RMSE|MAE | RMSE| MAE | RMSE| MAE|
> OTImputer|0.967|0.665|0.915|0.623 |0.944 |0.696|
> TDM| 0.960 | 0.732 | 0.976 | 0.740 | 0.997 |0.749 |
> CLWF|0.197| 0.128|0.263 |0.171 |0.323 |0.205|
>
> >2. Thanks for the suggestion. Please see the reponse to Reviewer cRw4.
> ### **Weaknesses**
> >1. no need for introducing concepts like Schrodinger bridge in the main content.
> #### **Response**: Thanks for the suggestion. The main reason we introduce the Schrodinger bridge problem (the dynamic optimal transport problem considering stochasticity) in the main text, is to link the sampling method to the SDE and SOC, which can help us provide a more generalized theoretical framework. We will move the relevant introduction to the appendix.
> >2.The detailed evaluation metric computational protocols should be included in the manuscript.
> #### **Response**: The RMSE: $RMSE = \sqrt{ \frac{1}{N} \sum_{i=1}^N (x_i-\hat{x}_i)^2}$, where $N$ is the total number of the imputation target datapoints, $x_i$ is the target data value, and .$\hat{x}_i$ is the imputation data value.
> #### The MAE: $MAE = \frac{1}{N} \sum_{i=1}^N |x_i-\hat{x}_i|$. We will include the computation protocols in the appendix.

---

> > ### Comment · Reviewer_n7gf · 2025-04-05
> >
> > It appears that the author's rebuttal has not fully addressed the reviewer's concerns. The specific points of contention are listed as follows:
> > 1. The acceleration mechanism for the DPM solver relies on the property that the OU process contains a linear drift term, allowing it to be modeled by an exponential integrator. However, the SDE presented in the manuscript does not include such a linear drift term. This suggests that the author may have misunderstood the underlying concepts.
> > 2. If the missing-data mechanism is not explicitly considered—for instance, under a Missing Not At Random (MNAR) scenario—this can lead to incorrect results. If this assumption is integral to the proposed approach, we recommend clearly articulating it in the manuscript.
> > 3. Please provide a detailed proof for Eq. (12) to substantiate the claim made in the proposal.
> > 4. The claim that strong friction can negate acceleration requires clarification. If this setup indeed renders related references irrelevant, could you explain why? Otherwise, it would suggest that those references [1-2] serve no meaningful purpose in the context of this discussion.
> > 5. The computation of MAE is given as \( MAE = \frac{1}{N} \sum_{i=1}^N |x_i - \hat{x}_i| \). However, based on `PyPOTS/pypots/nn/functional/error.py`, it seems that the mask matrix should be included in the calculation. Could you clarify this discrepancy?
> > 6. "If we can estimate the target distribution's mean unbiasedly, then a generative methodology is suitable for imputation tasks." However, when the target distribution is bimodal—such as a Gaussian Mixture Model (GMM)—which "mean" are we referring to?
> >
> > ---
> > References:
> > [1]. Accelerated Flow for Probability Distributions, ICML 2019
> > [2]. A variational perspective on accelerated methods in optimization, PNAS 2016

---

> > > ### Author Response · Authors · 2025-04-07
> > >
> > > Many thanks for the feedback.
> > > >**C1**: acceleration mechanism for the DPM solver...
> > >
> > > #### **R1**: The acceleration mechanism proposed the DPM solver aims to reduce the numerical error of the time integral for the semilinear probability flow ODE learned by a DDPM/VP-SDE (whose sampling path is not necessarily the optimal/shortest in the Wasserstein sense). In contrast, the drift term of the SDE learned by the OT theory and flow matching in the paper allows us to compute the geodesic/straight/shortest path in the Wasserstein space to accelerate the sampling speed. We will include the discussion in the paper.
> > >
> > > >**C2**: If the missing-data mechanism...
> > >
> > > #### **R2**: We are aware of the importance of the problem setup the reviewer raised. The MNAR scenario is indeed tricky to handle as it requires domain-specific knowledge. In our setting, the masks are generated independently of the data. This problem formulation is consistent with numerous prior works on time series imputation, as listed below. Nevertheless, we will explicitly clarify this issue in the paper.
> > > #### References:
> > > #### [1] Fortuin, et al. Gp-vae: Deep probabilistic time series imputation. AISTATS, 2020.
> > > #### [2] Tashiro, et al. Csdi: Conditional score-based diffusion models for probabilistic time series imputation. NeurIPS, 2021.
> > > #### [3] Bilos, et al. Modeling temporal data as continuous functions with stochastic process diffusion. ICML, 2023.
> > > #### [4] Wu, et al. Timesnet: Temporal 2d-variation modeling for general time series analysis. ICLR, 2023.
> > > #### [5] Chen, et al. Provably convergent Schrodinger bridge with applications to probabilistic time series imputation. ICML, 2023
> > > #### [6] Wang, et al. Optimal Transport for Time Series Imputation. ICLR, 2025.
> > >
> > > >**C3**: Please provide a detailed proof ...
> > >
> > > #### **R3**: We can understand the flow dynamics from two different perspectives, namely, the Lagrangian specification and the Eulerian specification. In the Lagrangian specification of the flow field, we focus on the motion of the induvial particles. This can be depicted by an SDE or Eq (12). In the Eulerian specification of the flow field, we focus on the time evolution of the population/probability density, which is depicted by the Fokker-Planck equation. If we are able to make $X_t$ follow the $\frac{dX_t}{dt} = \mu(X,t)$ and satisfies the density function $p(X_t)$ at the same time (that is what we are trying to do in the paper), then the corresponding Fokker-Planck equation is satisfied (please refer to [7]). In this way, we can further formulate the base sampler and the sufficient statistic in the Rao-Blackwellization sense. We will include the detailed explanation in the paper.
> > > #### Reference:
> > > #### [7] Liu. Rectified flow: A marginal preserving approach to optimal transport. arXiv preprint arXiv:2209.14577 (2022).
> > >
> > > >**C4**: The claim that strong friction...
> > > #### **R4**: In the context of Langevin dynamics, when the friction is very strong, the timescale of inertial motion becomes extremely short, and the system quickly reaches terminal velocity where friction balances the other forces. Therefore, the acceleration becomes negligible (please see [8-9]). Note that $\frac{dx_t}{dt} = -\nabla U (x_t)$ is also the strong-friction limit of $ \lambda \rightarrow \infty$ of the second-order ODE $\ddot{x}_t + \lambda \dot{x}_t + \lambda \nabla U (x_t) = 0$ in Nesterov’s acceleration method (please refer to [10], SI Appendix, H1). In our method, we do not estimate the kinetic energy and potential energy  at the same time to model the dynamics (unlike in [10-11]). Instead, we estimate the same flow dynamics from two different perspectives indepdently, $\frac{dx_t}{dt} =v_t(x,t)$ (related to the kinetic energy) and $\frac{dx_t}{dt} = -\nabla U(x_t)$ (related to the potential energy, which we also function can serve as the control signal, please see Appendix B4). This enables us to formulate a new Rao-Blackwellized sampler to improve the model’s performance.
> > > #### References:
> > > #### [8] Villani, Cédric. Optimal transport: old and new. Vol. 338. Berlin: springer, 2008, pp 646.
> > > #### [9] Bussi, et al. Accurate sampling using Langevin dynamics. Physical Review E, 2007.
> > > #### [10] A variational perspective on accelerated methods in optimization, PNAS 2016.
> > > #### [11]  Accelerated Flow for Probability Distributions, ICML 2019
> > >
> > > >**C5**: The computation of MAE...
> > > #### **R5**: Thanks for the comment. There is no discrepancy. We included the mask matrix in the calculation in exactly the same manner as in previous works (e.g., CSDI). We will clarify this in the paper.
> > >
> > > >**C6**: which "mean" are we referring to?
> > > #### **R6**: We do not consider the multimodal issue in the proposed approach, which makes the task numerically easier to solve. The mean we refer to is the marginal mean of the predictions, which we believe is a common practice in the literature as well. And this also is a reasonable choice, as it achieves the best empirical performance in terms of RMSE and MAE.

---

### Official Review · Reviewer_bMau · 2025-03-14

**Overall Recommendation:** 1

**Summary:**

The article proposes a methodology for time series imputation using Wasserstein flows. The paper presents a number of theoretical elements required for their contribution, to then validate their method via simulations.

## update after rebuttal
As I posted early in the discussion period, the rebuttal does not cover the concerns I raised. Therefore, I maintain my recommendation.

**Claims And Evidence:**

The paper starts (abstract and intro) by motivating their work with the drawbacks of diffusion models for data imputation, namely, computational complexity and slow convergence. However, their proposal is never assessed in these terms, but only in estimation performance (MSE).

**Essential References Not Discussed:**

no reference to non-diffusion interpolation methods, such as neural processes, transformer-based or graph-based methods

**Experimental Designs Or Analyses:**

see "methods and evaluations above"

**Methods And Evaluation Criteria:**

The presentation of their proposal is not clear. The paper starts revising the preliminaries (sec 2) to then present their methodology in two pages (from mid page 3 to mid page 5).  It is hard to identify what in this presentation is novel and what is just an application of known results. The lack of clarity also applies to their diagram in Fig 1, in the perspective of this reviewer, that diagram provides no clarification or insights into the paper proposal. Overall, and since the formulation of the problem at the beginning of the paper, it is very difficult to understand what the authors are doing.

Though the proposed method outperforms the benchmarks according to the experiments in Sec 4, the poor performance in the first toy example is contradictory - see fig 2. In this rather simple example, any interpolator (splines, GP, polynomial) will perform better than the shown results, which exhibit overshoot and nonstationarity of errors.

There is also a number of typos and unprecise statements, for instance, line 149 (right) states to "solve" Eq. (7) - however, note that Eq. (7) is a definition, how can it be "solved"?

**Other Comments Or Suggestions:**

no

**Other Strengths And Weaknesses:**

The paper presents a promising and interesting idea, but the presentation is vague to the point that it is difficult to understand how such ideas are implemented, which hinders its reproducibility

**Questions For Authors:**

please refer to my criticisms above.

**Relation To Broader Scientific Literature:**

This is a very active line of research (diffusion/probabilistic models for time series) and the paper recognises key works in the area.

**Theoretical Claims:**

There are no theorems or other results of that kind and the connection with known results is vague (e.g., Rao-Blackwellisation)

---

> ### Author Rebuttal · Authors · 2025-03-31
>
> ### **Methods And Evaluation Criteria**
> >  line 149 (right) states to "solve" Eq. (7) - however, note that Eq. (7) is a definition, how can it be "solved"?
> #### **Response**: In Eq. (7) $\mu_t$ is unkonwn.
>
> ### **Supplementary Material**
> >For instance, what is the point of Fig 3?
> #### **Response**: Fig. 3 illustrates the connections and transformations among the different mathematical methods used in the proposed approach.

---

> > ### Comment · Reviewer_bMau · 2025-04-02
> >
> > The rebuttal does not cover the concerns I raised (e.g., the claims of addressing computational complexity and slow convergence in DMs or the performance in Fig 2). Therefore, I maintain my recommendation.

---

> > > ### Author Response · Authors · 2025-04-07
> > >
> > > We kindly request the reviewer to read our paper and responses to the comments from the other reviewers carefully, and make a fair evlaution of our paper accordingly.

---

### Official Review · Reviewer_cRw4 · 2025-03-15

**Overall Recommendation:** 4

**Summary:**

This paper introduces Conditional Lagrangian Wasserstein Flow (CLWF), time series imputation model that leverages optimal transport theory and Lagrangian mechanics. Following the principle of least action, CLWF learns a velocity field by minimizing kinetic energy, effectively finding the shortest path in probability space to generate missing values​. Experiments on synthetic and real-world datasets show that CLWF achieves competitive imputation performance compared to state-of-the-art methods while requiring fewer sampling steps.

**Claims And Evidence:**

-  The authors claim that their method(CLWF) achieves competitive imputation performance compared to state-of-the-art methods while requiring fewer sampling steps, thereby offering faster inference than diffusion-based approaches​. This claim is well supported by extensive experiments.

- The authors claim that they show the connection between proposed method and SOC,path measures. But in my opinion, more supporting detail should be included. In this version of manuscript, I cannot find detailed relationship between CLWF and these concepts.

**Essential References Not Discussed:**

To the best of my knowledge, the authors properly cite relevant papers except for concurrent work [1] .



[1] Wang et al., "Optimal transport for time-series imputation" , ICLR 2025

**Experimental Designs Or Analyses:**

This work follows experimental designs from prior work in general. I think experimental designs are sound.

**Methods And Evaluation Criteria:**

I have checked that methods and evaluation criteria make sense for time series imputation.

**Other Comments Or Suggestions:**

Please refer to other sections.

**Other Strengths And Weaknesses:**

**Strengths**
- This paper is well written, easy to follow. Description of relationship among SOC,OT, flow matching in supplementary is helpful for broad audience.
- Timely topic, novel approach in time series imputation using flow based method, reducing cost of diffusion model for imputation.
- Decent performance on real world benchmark.

**Weaknesses**
- I think it would be better to discuss the difference and relationship between similar models like CSBI(e.g., including algorithm in supplementary for comparing actual difference in computation)

**Questions For Authors:**

- How this paper relates to the recently published Hao et al., "Optimal transport for time-series imputation" , ICLR 2025
- It would be interesting to apply this for forecasting or extraploation tasks.
- What is the benefit of framing this imputation problem within the framework of Lagrangian mechanics? What is difference between this method and directly applying flow matching method?
- I think including actual runtime for each method will be helpful.

**Relation To Broader Scientific Literature:**

This paper employs flow matching and optimal transport for time series imputation. This is new in this literature and will be valuable.

**Theoretical Claims:**

There are some theoretical concepts in this paper, but not including theorem specifically requires proof.

---

> ### Author Rebuttal · Authors · 2025-03-29
>
> Thank you very much for the insigtful comments.
> ### **Claims And Evidence**
> >1. The authors claim that they show the connection between proposed method and SOC,path measures. But in my opinion, more supporting detail should be included. In this version of manuscript, I cannot find detailed relationship between CLWF and these concepts.
> #### **Response**: In CLWF, we treat the data sampling process as a controlled SDE, which is the formulation of an SOC problem (Eq. 18 in Appendix B.1). And the control signal can be the energy function estimated by the TDVE, which is proved in Appendix B. The loss function of CLWF also minimize the cost function of the SOC problem (Eq. 19 in Appendix B.1). Further, the control function can be used as the sufficient statistic to formulate a new Rao-Blackwellized sampler (please refer to Appendix C) to improve the model’s performance. Finally, the control signal of the SOC framework formulates a new path measure (i.e., the sampling path is now controlled) according to the Girsanov theorem. Please refer to Appendix B.4 and Fig. 3 in Appendix C.1.
> >2. There are some theoretical concepts in this paper, but not including theorem specifically requires proof.
> #### **Response**: Thanks for the suggestion. We will add the detailed proof in the paper.
>
> ### **Essential References Not Discussed**
> >To the best of my knowledge, the authors properly cite relevant papers except for concurrent work [1].
> #### **Response**: Thanks for the suggestion. [1] proposed a novel specialized static OT metric which uses DFT to transform the original time series data into the frequency domain for imputation, while CLWF sloves the dynamical OT problem. We will discuss and cite the paper in the main text.
> ### **Weaknesses**
> >I think it would be better to discuss the difference and relationship between similar models like CSBI(e.g., including algorithm in supplementary for comparing actual difference in computation)
> #### **Response**: Thanks for the suggestion. Both CSBI and CLWF try to solve the dynamical OT/Schrodinger bridge problem. CSBI solves the SBP via the diffusion model and approximate iterative proportional fitting (IPF); while CLWF adopts the Lagrangian dynamics framework to solve the sampling problem using flow matching and further improve the model's performance via Rao-Blackwellization using a TDVE. We will discuss the difference and relationship between CLWF and similar methods in the main paper and give the detailed algorithm comparison in the appendix.
> ### **Questions For Authors**
> > How this paper relates to the recently published Hao et al., "Optimal transport for time-series imputation" , ICLR 2025
> #### **Response**: Please see above.
> > It would be interesting to apply this for forecasting or extraploation tasks.
> #### **Response**: Thanks for the suggestion. We will explore these tasks in future work.
> >What is the benefit of framing this imputation problem within the framework of Lagrangian mechanics? What is difference between this method and directly applying flow matching method?
> #### **Response**:Both Lagrangian mechanics and flow matching enables us obtain the shortest path for the dynamic optimal transport problem. However, compared to the flow matching method which only considers the drift/velocity, the framework of Lagrangian mechanics includes both the drift term and control term which is derived from the potential energy term, which as a result provides us a more general sampling framework. This enables us to formulate a Rao-Blackwellized sampler to further reduce the variances of the sampler; therefore, the performance of the base sampler (based on flow matching) is improved. Moreover; the framework of Lagrangian mechanics also enables us to bridge the gap between the optimal transport and SOC problem. Please also refer to the references listed below.
> ##### [1] Liu, G.-H., Lipman, Y., Nickel, M., Karrer, B., Theodorou, E., and Chen, R. T. Generalized Schrodinger bridge matching. In The Twelfth International Conference on Learning Representations, 2024.
>
> ##### [2] Koshizuka, T. and Sato, I. Neural Lagrangian Schrodinger bridge: Diffusion modeling for population dynamics. In The Eleventh International Conference on Learning Representations, 2023.
>
> ##### [3] Neklyudov, K., Brekelmans, R., Severo, D., and Makhzani, A. Action matching: Learning stochastic dynamics from samples. In International Conference on Machine Learning, pp. 25858–25889. PMLR, 2023a.
>
> ##### [4] Neklyudov, K., Brekelmans, R., Tong, A., Atanackovic, L., Liu, Q., and Makhzani, A. A computational framework for solving wasserstein Lagrangian flows. arXiv preprint arXiv:2310.10649, 2023b
>
> >I think including actual runtime for each method will be helpful.
> #### **Response**:   Thanks for the suggestion. We show the actual runtime below.
> Method| CSDI | DSPD-GP| Base | RB | Resampling | Resampling + RB |
> --- | --- | --- |---- |---- | --- |---- |
> runtime/s|  354.65   | 356.82|141.86 |276.75 | 142.73       |277.62

---

> > ### Comment · Reviewer_cRw4 · 2025-04-06
> >
> > Thanks for answering to my questions and giving additional experiments. I am happy to raise my score since this makes actual gain in the performance

---

> > > ### Author Response · Authors · 2025-04-06
> > >
> > > We are glad that our response addressed your questions. Thank you very much for raising the score!

---

### Decision · Program_Chairs · 2025-05-01

**Decision:**

Reject

**Comment:**

The paper introduces a new method for time-series imputation that reformulates the problem as an optimization of a potential function.
While the paper introduces an innovative approach, reviewers found that there are several concerns about the work
that have not been clarified by the rebuttals and hence suggest rejection of the paper.
The key points are the lack of motivation and its limited scope (not addressing the MNAR scenario).